# VILD: Variational Imitation Learning with Diverse-quality Demonstrations

## Abstract

The goal of imitation learning (IL) is to learn a good policy from high-quality demonstrations. However, the quality of demonstrations in reality can be diverse, since it is easier and cheaper to collect demonstrations from a mix of experts and amateurs. IL in such situations can be challenging, especially when the level of demonstrators' expertise is unknown. We propose a new IL paradigm called Variational Imitation Learning with Diverse-quality demonstrations (VILD), where we explicitly model the level of demonstrators' expertise with a probabilistic graphical model and estimate it along with a reward function. We show that a naive estimation approach is not suitable to large state and action spaces, and fix this issue by using a variational approach that can be easily implemented using existing reinforcement-learning methods. Experiments on continuous-control benchmarks and real-world crowdsourced demonstrations denote that VILD outperforms state-of-the-art methods. Our work enables scalable and data-efficient IL under more realistic settings than before.

## 1 Introduction

The goal of sequential decision making is to learn a policy that makes good decisions (Puterman, 1994). As an important branch of sequential decision making, imitation learning (IL) (Schaal, 1999) aims to learn such a policy from demonstrations (i.e., sequences of decisions) collected from experts. However, high-quality demonstrations can be difficult to obtain in reality, since such experts may not always be available and sometimes are too costly (Osa et al., 2018). This is especially true when the quality of decisions depends on specific domain-knowledge not typically available to amateurs; e.g., in applications such as robot control (Osa et al., 2018) and autonomous driving (Silver et al., 2012).

In practice, demonstrations are often diverse in quality, since it is cheaper to collect demonstrations from mixed demonstrators, containing both experts and amateurs (Audiffren et al., 2015). Unfortunately, IL in such settings tends to perform poorly, since low-quality demonstrations often negatively affect the performance of IL methods (Shiarlis et al., 2016). For example, amateurs' demonstrations for robotics can be cheaply collected via a robot simulation (Mandlekar et al., 2018), but such demonstrations may cause damages to the robot which is catastrophic in the real-world (Shiarlis et al., 2016). Similarly, demonstrations for autonomous driving can be collected from drivers in public roads (Fridman et al., 2017), which may contain traffic-accident demonstrations. Learning a self-driving car from these low-quality demonstrations may cause traffic accidents.

When the level of demonstrators' expertise is known, multi-modal IL (MM-IL) can be used to learn a good policy with diverse-quality demonstrations (Li et al., 2017; Hausman et al., 2017; Wang et al., 2017). Specifically, MM-IL aims to learn a multi-modal policy, where each mode of the policy represents the decision making of each demonstrator. When knowing the level of demonstrators' expertise, good policies can be obtained by selecting modes that correspond to the decision making of high-expertise demonstrators. However, in practice, it is difficult to truly determine the level of demonstrators' expertise beforehand. Without knowing the level of expertise, it is difficult to distinguish the decision making of experts and amateurs, and learning a good policy is challenging.

To overcome the issue of MM-IL, pioneer works have proposed to estimate the quality of each demonstration using auxiliary information from experts (Audiffren et al., 2015; Wu et al., 2019; Brown et al., 2019). Specifically, Audiffren et al. (2015) inferred the demonstration quality using

similarities between diverse-quality demonstrations and high-quality demonstrations, where the latter are collected in a small number from experts. In contrast, Wu et al. (2019) proposed to estimate the demonstration quality using a small number of demonstrations with confidence scores. Namely, the score value given by an expert is proportion to the demonstration quality. Similarly, the demonstration quality can be estimated by ranked demonstrations, where ranking from an expert is evaluated due to the relative quality (Brown et al., 2019). To sum up, these methods rely on auxiliary information from experts, namely high-quality demonstrations, confidence scores, and ranking. In practice, these pieces of information can be scarce or noisy, which leads to a poor performance of these methods.

In this paper, we consider a novel but realistic setting of IL where only diverse-quality demonstrations are available. Meanwhile, the level of demonstrators' expertise and auxiliary information from experts are fully absent. To tackle this challenging setting, we propose a new learning paradigm called variational imitation learning with diverse-quality demonstrations (VILD). The central idea of VILD is to model the level of demonstrators' expertise via a probabilistic graphical model, and learn it along with a reward function that represents an intention of expert's decision making. To scale up our model for large state and action spaces, we leverage the variational approach (Jordan et al., 1999), which can be implemented using reinforcement learning (RL) for flexibility (Sutton & Barto, 1998). To further improve data-efficiency of VILD when learning the reward function, we utilize importance sampling (IS) to re-weight a sampling distribution according to the estimated level of demonstrators' expertise. Experiments on continuous-control benchmarks and real-world crowdsourced demonstrations (Mandlekar et al., 2018) denote that: 1) VILD is robust against diverse-quality demonstrations and outperforms existing methods significantly. 2) VILD with IS is data-efficient, since it learns the policy using a less number of transition samples.

## 2    IL FROM DIVERSE-QUALITY DEMONSTRATIONS AND ITS CHALLENGE

Before delving into our main contribution, we first give the minimum background about RL and IL. Then, we formulate a new setting in IL called *diverse-quality demonstrations*, discuss its challenge, and reveal the deficiency of existing methods.

**Reinforcement learning.**    Reinforcement learning (RL) (Sutton & Barto, 1998) aims to learn an optimal policy of a sequential decision making problem, which is often mathematically formulated as a Markov decision process (MDP) (Puterman, 1994). We consider a finite-horizon MDP with continuous state and action spaces defined by a tuple $\mathcal{M} = (\mathcal{S}, \mathcal{A}, p(\mathbf{s}'|\mathbf{s}, \mathbf{a}), p_1(\mathbf{s}_1), r(\mathbf{s}, \mathbf{a}))$ with a state $\mathbf{s}_t \in \mathcal{S} \subseteq \mathbb{R}^{d_\mathbf{s}}$, an action $\mathbf{a}_t \in \mathcal{A} \subseteq \mathbb{R}^{d_\mathbf{a}}$, an initial state density $p_1(\mathbf{s}_1)$, a transition probability density $p(\mathbf{s}_{t+1}|\mathbf{s}_t, \mathbf{a}_t)$, and a reward function $r : \mathcal{S} \times \mathcal{A} \mapsto \mathbb{R}$, where the subscript $t \in \{1, \ldots, T\}$ denotes the time step. A sequence of states and actions, $(\mathbf{s}_{1:T}, \mathbf{a}_{1:T})$, is called a trajectory. A decision making of an agent is determined by a policy $\pi(\mathbf{a}_t|\mathbf{s}_t)$, which is a conditional probability density of action given state. RL seeks for an optimal policy $\pi^\star(\mathbf{a}_t|\mathbf{s}_t)$ which maximizes the expected cumulative reward: $\mathbb{E}_{p_\pi(\mathbf{s}_{1:T}, \mathbf{a}_{1:T})}[\Sigma_{t=1}^T r(\mathbf{s}_t, \mathbf{a}_t)]$, where $p_\pi(\mathbf{s}_{1:T}, \mathbf{a}_{1:T}) = p_1(\mathbf{s}_1)\Pi_{t=1}^T p(\mathbf{s}_{t+1}|\mathbf{s}_t, \mathbf{a}_t)\pi(\mathbf{a}_t|\mathbf{s}_t)$ is a trajectory probability density induced by $\pi$. RL has shown great successes recently, especially when combined with deep neural networks (Silver et al., 2017). However, a major limitation of RL is that it relies on the reward function which may be unavailable in practice (Schaal, 1999).

**Imitation learning.**    To address the above limitation of RL, imitation learning (IL) was proposed (Schaal, 1999). Without using the reward function, IL aims to learn the optimal policy from demonstrations that encode information about the optimal policy. A common assumption in most IL methods is that, demonstrations are collected by $K \geqslant 1$ demonstrators who execute actions $\mathbf{a}_t$ drawn from $\pi^\star(\mathbf{a}_t|\mathbf{s}_t)$ for every states $\mathbf{s}_t$. A graphical model describing this data collection process is depicted in Figure 1(a), where a random variable $k \in \{1, \ldots, K\}$ denotes each demonstrator's identification number and $p(k)$ denotes the probability of collecting a demonstration from the $k$-th demonstrator. Under this assumption, demonstrations $\{(\mathbf{s}_{1:T}, \mathbf{a}_{1:T}, k)_n\}_{n=1}^N$ (i.e., observed random variables in Figure 1(a)) are called *expert demonstrations* and are regarded to be drawn independently from a probability density $p^\star(\mathbf{s}_{1:T}, \mathbf{a}_{1:T})p(k) = p(k)p_1(\mathbf{s}_1)\Pi_{t=1}^T p(\mathbf{s}_{t+1}|\mathbf{s}_t, \mathbf{a}_t)\pi^\star(\mathbf{a}_t|\mathbf{s}_t)$. We note that $k$ does not affect the trajectory density $p^\star(\mathbf{s}_{1:T}, \mathbf{a}_{1:T})$ and can be omitted. We assume a common assumption that $p_1(\mathbf{s}_1)$ and $p(\mathbf{s}_{t+1}|\mathbf{s}_t, \mathbf{a}_t)$ are unknown but we can sample states from them.

IL has shown great successes in benchmark settings (Ho & Ermon, 2016; Fu et al., 2018; Peng et al., 2019). However, practical applications of IL in the real-world is relatively few (Schroecker et al., 2019). One of the main reasons is that most IL methods aim to learn with expert demonstrations. In practice, such demonstrations are often too costly to obtain due to a limited number of experts, and

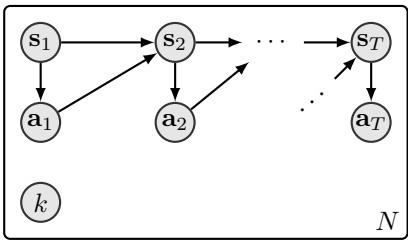
(a) Expert demonstrations.

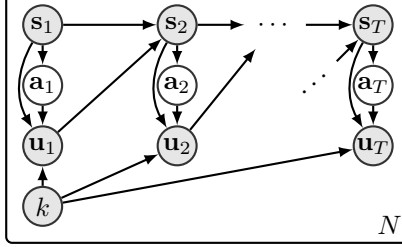
(b) Diverse-quality demonstrations.

Figure 1: Graphical models describe expert demonstrations and diverse-quality demonstrations. Shaded and unshaded nodes indicate observed and unobserved random variables, respectively. Plate notations indicate that the sampling process is repeated for $N$ times. $\mathbf{s}_t \in \mathcal{S}$ is a state with transition densities $p(\mathbf{s}_{t+1}|\mathbf{s}_t, \mathbf{a}_t)$, $\mathbf{a}_t \in \mathcal{A}$ is an action with density $\pi^\star(\mathbf{a}_t|\mathbf{s}_t)$, $\mathbf{u}_t \in \mathcal{A}$ is a noisy action with density $p(\mathbf{u}_t|\mathbf{s}_t, \mathbf{a}_t, k)$, and $k \in \{1, \ldots, K\}$ is an identification number with distribution $p(k)$.

even when we obtain them, the number of demonstrations is often too few to accurately learn the optimal policy (Audiffren et al., 2015; Wu et al., 2019; Brown et al., 2019).

**New setting in IL: Diverse-quality demonstrations.** To improve practicality of IL, we consider a new learning paradigm called *IL with diverse-quality demonstrations*, where demonstrations are collected from demonstrators with different level of expertise. Compared to expert demonstrations, diverse-quality demonstrations can be collected more cheaply, e.g., via crowdsourcing (Mandlekar et al., 2018). The graphical model in Figure 1(b) depicts the process of collecting such demonstrations from $K > 1$ demonstrators. Formally, we select the $k$-th demonstrator according to a distribution $p(k)$. After selecting $k$, for each time step $t$, the $k$-th demonstrator observes state $\mathbf{s}_t$ and samples action $\mathbf{a}_t$ using $\pi^\star(\mathbf{a}_t|\mathbf{s}_t)$. However, the demonstrator may not execute $\mathbf{a}_t$ in the MDP if this demonstrator is not expertised. Instead, he/she may sample an action $\mathbf{u}_t \in \mathcal{A}$ with another probability density $p(\mathbf{u}_t|\mathbf{s}_t, \mathbf{a}_t, k)$ and execute it. Then, the next state $\mathbf{s}_{t+1}$ is observed with a probability density $p(\mathbf{s}_{t+1}|\mathbf{s}_t, \mathbf{u}_t)$, and the demonstrator continues making decision until time step $T$. We repeat this process for $N$ times to collect *diverse-quality demonstrations* $\mathcal{D}_\mathrm{d} = \{(\mathbf{s}_{1:T}, \mathbf{u}_{1:T}, k)_n\}_{n=1}^N$. These demonstrations are regarded to be drawn independently from a probability density

$$p_\mathrm{d}(\mathbf{s}_{1:T}, \mathbf{u}_{1:T}|k)p(k) = p(k)p(\mathbf{s}_1)\prod_{t=1}^T p_1(\mathbf{s}_{t+1}|\mathbf{s}_t, \mathbf{u}_t)\int_{\mathcal{A}} \pi^\star(\mathbf{a}_t|\mathbf{s}_t)p(\mathbf{u}_t|\mathbf{s}_t, \mathbf{a}_t, k)\mathrm{d}\mathbf{a}_t. \quad (1)$$

We refer to $p(\mathbf{u}_t|\mathbf{s}_t, \mathbf{a}_t, k)$ as a noisy policy of the $k$-th demonstrator, since it is used to execute a noisy action $\mathbf{u}_t$. Our goal is to learn the optimal policy $\pi^\star$ using diverse-quality demonstrations $\mathcal{D}_\mathrm{d}$.

Note that Eq. (1) can be described equivalently by using a marginal density $\pi(\mathbf{u}_t|\mathbf{s}_t, k) = \int_{\mathcal{A}} \pi^\star(\mathbf{a}_t|\mathbf{s}_t)p(\mathbf{u}_t|\mathbf{s}_t, \mathbf{a}_t, k)\mathrm{d}\mathbf{a}_t$ and removing $\mathbf{a}_t$ from the graphical model. However, we explicitly write $\mathbf{a}_t$ as above to emphasize the dependency between $\pi^\star(\mathbf{a}_t|\mathbf{s}_t)$ and $p(\mathbf{u}_t|\mathbf{s}_t, \mathbf{a}_t, k)$. This emphasis will be made more clear in Section 3.1 when we describe our choice of model.

**The deficiency of existing methods.** We conjecture that existing IL methods are not suitable to learn with diverse-quality demonstrations according to $p_\mathrm{d}$. Specifically, these methods always treat observed demonstrations as if they were drawn from $p^\star$. By comparing $p^\star$ and $p_\mathrm{d}$, we can see that existing methods would learn $\pi(\mathbf{u}_t|\mathbf{s}_t)$ such that $\pi(\mathbf{u}_t|\mathbf{s}_t) \approx \Sigma_{k=1}^K p(k)\int_{\mathcal{A}} \pi^\star(\mathbf{a}_t|\mathbf{s}_t)p(\mathbf{u}_t|\mathbf{s}_t, \mathbf{a}_t, k)\mathrm{d}\mathbf{a}_t$. In other words, they learn a policy that averages over decisions of all demonstrators. This would be problematic when amateurs are present, as averaged decisions of all demonstrators would be highly different from those of all experts. Worse yet, state distributions of amateurs and experts tend to be highly different, which often leads to the unstable learning: The learned policy oscillated between well-performed policy and poorly-performed policy. For these reasons, we believe that existing methods tend to learn a policy that achieves average performances, and are not suitable for handling the setting of diverse-quality demonstrations.

# 3 VILD: A ROBUST METHOD FOR DIVERSE-QUALITY DEMONSTRATIONS

This section presents VILD, namely a robust method for tackling the challenge from diverse-quality demonstrations. Specifically, we build a probabilistic model that explicitly describes the level of demonstrators' expertise and a reward function (Section 3.1), and estimate its parameters by a variational approach (Section 3.2), which can be implemented easily by RL (Section 3.3). We also improve data-efficiency by using importance sampling (Section 3.4). Mathematical derivations are provided in Appendix A.

## 3.1 MODELING DIVERSE-QUALITY DEMONSTRATIONS

This section describes a model which enables estimating the level of demonstrators' expertise. We first describe a naive model, whose parameters can be estimated trivially via supervised learning, but suffers from the issue of compounding error. Then, we describe our proposed model, which avoids the issue of the naive model by learning a reward function.

**Naive model.** Based on $p_\mathrm{d}$, one of the simplest models to handle diverse-quality demonstrations is $p_{\boldsymbol{\theta},\boldsymbol{\omega}}(\mathbf{s}_{1:T}, \mathbf{u}_{1:T}, k) = p(k)p(\mathbf{s}_1)\Pi_{t=1}^{T}p(\mathbf{s}_{t+1}|\mathbf{s}_t, \mathbf{u}_t)\int_{\mathcal{A}} \pi_{\boldsymbol{\theta}}(\mathbf{a}_t|\mathbf{s}_t)p_{\boldsymbol{\omega}}(\mathbf{u}_t|\mathbf{s}_t, \mathbf{a}_t, k)\mathrm{d}\mathbf{a}_t$, where $\pi_{\boldsymbol{\theta}}$ and $p_{\boldsymbol{\omega}}$ are learned to estimate the optimal policy and the noisy policy, respectively. The parameters $\boldsymbol{\theta}$ and $\boldsymbol{\omega}$ can be learned by minimizing the Kullback-Leibler (KL) divergence from the data distribution to the model. This naive model can be regarded as an extension of a model proposed by Raykar et al. (2010) for handling diverse-quality data in supervised learning.

The main advantage of this naive model is that its parameters can be estimated trivially via supervised learning. However, this native model suffers from the issue of compounding error (Ross & Bagnell, 2010) and tends to perform poorly. Specifically, supervised-learning methods assume that data distributions during training and testing are identical. However, data distributions during training and testing are different in IL, since data distributions depend on policies (Puterman, 1994). A discrepancy of data distributions causes compounding errors during testing, where prediction errors increase further in future predictions. Due to this issue, supervised-learning methods often perform poorly in IL (Ross & Bagnell, 2010). The issue becomes even worse with diverse-quality demonstrations, since data distributions of different demonstrators tend to be highly different. For these reasons, this naive model is not suitable for our setting.

**Proposed model.** To avoid the issue of compounding error, our method utilizes the inverse RL (IRL) approach (Ng & Russell, 2000), where we aim to learn a reward function from diverse-quality demonstrations[1]. IL problems can be solved by a combination of IRL and RL, where we learn a reward function by IRL and then learn a policy from the reward function by RL. This combination avoids the issue of compounding error, since the policy is learned by RL which generalizes to states not presented in demonstrations (Ho & Ermon, 2016).

Specifically, our proposed model is based on a model of maximum entropy IRL (MaxEnt-IRL) (Ziebart et al., 2010). Briefly speaking, MaxEnt-IRL learns a reward function from expert demonstrations by using a model $p_{\boldsymbol{\phi}}(\mathbf{s}_{1:T}, \mathbf{a}_{1:T}) \propto p(\mathbf{s}_1)\Pi_{t=1}^{T}p_1(\mathbf{s}_{t+1}|\mathbf{s}_t, \mathbf{a}_t) \exp(r_{\boldsymbol{\phi}}(\mathbf{s}_t, \mathbf{a}_t))$. Based on this model, we propose to learn the reward function *and* the level of expertise by a model

$$p_{\boldsymbol{\phi},\boldsymbol{\omega}}(\mathbf{s}_{1:T}, \mathbf{u}_{1:T}, k) \propto p(k)p_1(\mathbf{s}_1) \prod_{t=1}^{T} p(\mathbf{s}_{t+1}|\mathbf{s}_t, \mathbf{u}_t) \int_{\mathcal{A}} \exp\left(r_{\boldsymbol{\phi}}(\mathbf{s}_t, \mathbf{a}_t)\right) p_{\boldsymbol{\omega}}(\mathbf{u}_t|\mathbf{s}_t, \mathbf{a}_t, k)\mathrm{d}\mathbf{a}_t, \quad (2)$$

where $\boldsymbol{\phi}$ and $\boldsymbol{\omega}$ are parameters. We denote a normalization term of this model by $Z_{\boldsymbol{\phi},\boldsymbol{\omega}}$. By comparing the proposed model $p_{\boldsymbol{\phi},\boldsymbol{\omega}}$ to the data distribution $p_\mathrm{d}$, the reward parameter $\boldsymbol{\phi}$ should be learned so that the cumulative reward is proportion to a joint probability density of actions given by the optimal policy, i.e., $\exp(\Sigma_{t=1}^{T}r_{\boldsymbol{\phi}}(\mathbf{s}_t, \mathbf{a}_t)) \propto \Pi_{t=1}^{T}\pi^{\star}(\mathbf{a}_t|\mathbf{s}_t)$. In other words, the cumulative reward is large for trajectories induced by the optimal policy. Therefore, the optimal policy can be learned by maximizing the cumulative reward. Meanwhile, the density $p_{\boldsymbol{\omega}}(\mathbf{u}_t|\mathbf{s}_t, \mathbf{a}_t, k)$ is learned to estimate the noisy policy $p(\mathbf{u}_t|\mathbf{s}_t, \mathbf{a}_t, k)$. In the remainder, we refer to $\boldsymbol{\omega}$ as an expertise parameter.

To learn parameters of this model, we propose to minimize the KL divergence from the data distribution to the model: $\min_{\boldsymbol{\phi},\boldsymbol{\omega}} \mathrm{KL}(p_\mathrm{d}(\mathbf{s}_{1:T}, \mathbf{u}_{1:T}|k)p(k)||p_{\boldsymbol{\phi},\boldsymbol{\omega}}(\mathbf{s}_{1:T}, \mathbf{u}_{1:T}, k))$. By

---

[1]We emphasize that IRL is different from RL; IRL learns a reward function from demonstrations, whereas RL learns an optimal policy from a known reward function.

rearranging terms and ignoring constant terms, minimizing this KL divergence is equivalent to solving an optimization problem $\max_{\phi,\omega} f(\phi,\omega) - g(\phi,\omega)$, where $f(\phi,\omega) = \mathbb{E}_{p_d(\mathbf{s}_{1:T},\mathbf{u}_{1:T}|k)p(k)}[\Sigma_{t=1}^{T}\log(\int_{\mathcal{A}}\exp(r_\phi(\mathbf{s}_t,\mathbf{a}_t))p_\omega(\mathbf{u}_t|\mathbf{s}_t,\mathbf{a}_t,k)d\mathbf{a}_t)]$ and $g(\phi,\omega) = \log Z_{\phi,\omega}$. To solve this optimization, we need to compute the integrals over both state space $\mathcal{S}$ and action space $\mathcal{A}$. Computing these integrals is feasible for small state and action spaces, but is infeasible for large state and action spaces. To scale up our model to MDPs with large state and action spaces, we leverage a variational approach in the followings.

## 3.2 VARIATIONAL APPROACH FOR PARAMETER ESTIMATION

The central idea of the variational approach is to lower-bound an integral by the Jensen inequality and a variational distribution (Jordan et al., 1999). The main benefit of the variational approach is that the integral can be indirectly computed via the lower-bound, given an optimal variational distribution. However, finding the optimal distribution often requires solving a sub-optimization problem.

Before we proceed, notice that $f(\phi,\omega) - g(\phi,\omega)$ is not a joint concave function of the integrals, and this prohibits using the Jensen inequality. However, we can separately lower-bound $f$ and $g$ by the Jensen inequality, since they are concave functions of their corresponding integrals. Specifically, let $l_{\phi,\omega}(\mathbf{s}_t,\mathbf{a}_t,\mathbf{u}_t,k) = r_\phi(\mathbf{s}_t,\mathbf{a}_t) + \log p_\omega(\mathbf{u}_t|\mathbf{s}_t,\mathbf{a}_t,k)$. By using a variational distribution $q_\psi(\mathbf{a}_t|\mathbf{s}_t,\mathbf{u}_t,k)$ with parameter $\psi$, we obtain an inequality $f(\phi,\omega) \geqslant \mathcal{F}(\phi,\omega,\psi)$, where

$$\mathcal{F}(\phi,\omega,\psi) = \mathbb{E}_{p_d(\mathbf{s}_{1:T},\mathbf{u}_{1:T}|k)p(k)}\left[\Sigma_{t=1}^{T}\mathbb{E}_{q_\psi(\mathbf{a}_t|\mathbf{s}_t,\mathbf{u}_t,k)}\left[l_{\phi,\omega}(\mathbf{s}_t,\mathbf{a}_t,\mathbf{u}_t,k)\right] + H_t(q_\psi)\right], \quad (3)$$

and $H_t(q_\psi) = -\mathbb{E}_{q_\psi(\mathbf{a}_t|\mathbf{s}_t,\mathbf{u}_t,k)}\left[\log q_\psi(\mathbf{a}_t|\mathbf{s}_t,\mathbf{u}_t,k)\right]$. It is trivial to verify that the equality $f(\phi,\omega) = \max_\psi \mathcal{F}(\phi,\omega,\psi)$ holds (Jordan et al., 1999), where the maximizer $\psi^\star$ of the lower-bound yields $q_{\psi^\star}(\mathbf{a}_t|\mathbf{s}_t,\mathbf{u}_t,k) \propto \exp(l_{\phi,\omega}(\mathbf{s}_t,\mathbf{a}_t,\mathbf{u}_t,k))$. Therefore, the function $f(\phi,\omega)$ can be substituted by $\max_\psi \mathcal{F}(\phi,\omega,\psi)$. Meanwhile, by using a variational distribution $q_\theta(\mathbf{a}_t,\mathbf{u}_t|\mathbf{s}_t,k)$ with parameter $\theta$, we obtain an inequality $g(\phi,\omega) \geqslant \mathcal{G}(\phi,\omega,\theta)$, where

$$\mathcal{G}(\phi,\omega,\theta) = \mathbb{E}_{\widetilde{q}_\theta(\mathbf{s}_{1:T},\mathbf{u}_{1:T},\mathbf{a}_{1:T},k)}\left[\Sigma_{t=1}^{T}l_{\phi,\omega}(\mathbf{s}_t,\mathbf{a}_t,\mathbf{u}_t,k) - \log q_\theta(\mathbf{a}_t,\mathbf{u}_t|\mathbf{s}_t,k)\right], \quad (4)$$

and $\widetilde{q}_\theta(\mathbf{s}_{1:T},\mathbf{u}_{1:T},\mathbf{a}_{1:T},k) = p(k)p_1(\mathbf{s}_1)\Pi_{t=1}^{T}p(\mathbf{s}_{t+1}|\mathbf{s}_t,\mathbf{u}_t)q_\theta(\mathbf{a}_t,\mathbf{u}_t|\mathbf{s}_t,k)$. The lower-bound $\mathcal{G}$ resembles an objective function of maximum entropy RL (MaxEnt-RL) (Ziebart et al., 2010). By using the optimality results of MaxEnt-RL (Haarnoja et al., 2018), we have an equality $g(\phi,\omega) = \max_\theta \mathcal{G}(\phi,\omega,\theta)$. Therefore, the function $g(\phi,\omega)$ can be substituted by $\max_\theta \mathcal{G}(\phi,\omega,\theta)$.

By using these lower-bounds, we have that $\max_{\phi,\omega} f(\phi,\omega) - g(\phi,\omega) = \max_{\phi,\omega,\psi} \mathcal{F}(\phi,\omega,\psi) - \max_\theta \mathcal{G}(\phi,\omega,\theta) = \max_{\phi,\omega,\psi} \min_\theta \mathcal{F}(\phi,\omega,\psi) - \mathcal{G}(\phi,\omega,\theta)$. Solving the max-min problem is often feasible even for large state and action spaces, since $\mathcal{F}(\phi,\omega,\psi)$ and $\mathcal{G}(\phi,\omega,\theta)$ are defined as an expectation and can be optimized straightforwardly. Nevertheless, in practice, we represent the variational distributions by parameterized functions, and iteratively solve the sub-optimization (w.r.t. $\psi$ and $\theta$) by stochastic optimization methods. However, in this scenario, the equalities $f(\phi,\omega) = \max_\psi \mathcal{F}(\phi,\omega,\psi)$ and $g(\phi,\omega) = \max_\theta \mathcal{G}(\phi,\omega,\theta)$ may not hold for two reasons. First, the optimal variational distributions may not be in the space of our parameterized functions. Second, stochastic optimization methods may yield local solutions. Nonetheless, when the variational distributions are represented by deep neural networks, the obtained variational distributions are often reasonably accurate and the equalities approximately hold (Ranganath et al., 2014).

## 3.3 MODEL SPECIFICATION

In practice, we are required to specify models for $q_\theta$ and $p_\omega$. We propose to use $q_\theta(\mathbf{a}_t,\mathbf{u}_t|\mathbf{s}_t,k) = q_\theta(\mathbf{a}_t|\mathbf{s}_t)\mathcal{N}(\mathbf{u}_t|\mathbf{a}_t,\mathbf{\Sigma})$ and $p_\omega(\mathbf{u}_t|\mathbf{s}_t,\mathbf{a}_t,k) = \mathcal{N}(\mathbf{u}_t|\mathbf{a}_t,\mathbf{C}_\omega(k))$. As shown below, the choice for $q_\theta(\mathbf{a}_t,\mathbf{u}_t|\mathbf{s}_t,k)$ enables us to solve the sub-optimization w.r.t. $\theta$ by using RL with reward function $r_\phi$. Meanwhile, the choice for $p_\omega(\mathbf{u}_t|\mathbf{s}_t,\mathbf{a}_t,k)$ incorporates our prior assumption that the noisy policy tends to Gaussian, which is a reasonable assumption for actual human motor behavior (van Beers et al., 2004). Under these model specifications, solving $\max_{\phi,\omega,\psi} \min_\theta \mathcal{F}(\phi,\omega,\psi) - \mathcal{G}(\phi,\omega,\theta)$ is equivalent to solving $\max_{\phi,\omega,\psi} \min_\theta \mathcal{H}(\phi,\omega,\psi,\theta)$, where

$$\mathcal{H}(\phi,\omega,\psi,\theta) = \mathbb{E}_{p_d(\mathbf{s}_{1:T},\mathbf{u}_{1:T}|k)p(k)}\left[\Sigma_{t=1}^{T}\mathbb{E}_{q_\psi(\mathbf{a}_t|\mathbf{s}_t,\mathbf{u}_t,k)}\left[r_\phi(\mathbf{s}_t,\mathbf{a}_t) - \tfrac{1}{2}\|\mathbf{u}_t - \mathbf{a}_t\|_{\mathbf{C}_\omega^{-1}(k)}^2\right] + H_t(q_\psi)\right]$$

$$- \mathbb{E}_{\widetilde{q}_\theta(\mathbf{s}_{1:T},\mathbf{a}_{1:T})}\left[\Sigma_{t=1}^{T}r_\phi(\mathbf{s}_t,\mathbf{a}_t) - \log q_\theta(\mathbf{a}_t|\mathbf{s}_t)\right] + \frac{T}{2}\mathbb{E}_{p(k)}\left[\text{Tr}(\mathbf{C}_\omega^{-1}(k)\mathbf{\Sigma})\right]. \quad (5)$$

Here, $\tilde{q}_{\boldsymbol{\theta}}(\mathbf{s}_{1:T}, \mathbf{a}_{1:T}) = p_1(\mathbf{s}_1)\Pi_{t=1}^{T} \int_{\mathcal{A}} p(\mathbf{s}_{t+1}|\mathbf{s}_t, \mathbf{u}_t)\mathcal{N}(\mathbf{u}_t|\mathbf{a}_t, \boldsymbol{\Sigma})d\mathbf{u}_t q_{\boldsymbol{\theta}}(\mathbf{a}_t|\mathbf{s}_t)$ is a noisy trajectory density induced by a policy $q_{\boldsymbol{\theta}}(\mathbf{a}_t|\mathbf{s}_t)$, where $\mathcal{N}(\mathbf{u}_t|\mathbf{a}_t, \boldsymbol{\Sigma})$ can be regarded as an approximation of the noisy policy in Figure 1(b). Minimizing $\mathcal{H}$ w.r.t. $\boldsymbol{\theta}$ resembles solving a MaxEnt-RL problem with a reward function $r_{\boldsymbol{\phi}}(\mathbf{s}_t, \mathbf{a}_t)$, except that trajectories are collected according to the noisy trajectory density. In other words, this minimization problem can be solved using RL, and $q_{\boldsymbol{\theta}}(\mathbf{a}_t|\mathbf{s}_t)$ can be regarded as an approximation of the optimal policy. The hyper-parameter $\boldsymbol{\Sigma}$ determines the quality of this approximation: smaller value of $\boldsymbol{\Sigma}$ gives a better approximation. Therefore, by choosing a reasonably small value of $\boldsymbol{\Sigma}$, solving the max-min problem in Eq. (5) yields a reward function $r_{\boldsymbol{\phi}}(\mathbf{s}_t, \mathbf{a}_t)$ and a policy $q_{\boldsymbol{\theta}}(\mathbf{a}_t|\mathbf{s}_t)$. This policy imitates the optimal policy, which is the goal of IL.

The model specification for $p_{\boldsymbol{\omega}}$ incorporates our prior assumption about the noisy policy. Namely, $p_{\boldsymbol{\omega}}(\mathbf{u}_t|\mathbf{s}_t, \mathbf{a}_t, k) = \mathcal{N}(\mathbf{u}_t|\mathbf{a}_t, \mathbf{C}_{\boldsymbol{\omega}}(k))$ assumes that the noisy policy tends to Gaussian, where $\mathbf{C}_{\boldsymbol{\omega}}(k)$ gives an estimated expertise of the $k$-th demonstrator: High-expertise demonstrators have small $\mathbf{C}_{\boldsymbol{\omega}}(k)$ and vice-versa for low-expertise demonstrators. Note that VILD is not restricted to this choice. Different choices of $p_{\boldsymbol{\omega}}$ incorporate different prior assumptions. For example, a Laplace distribution incorporates a prior assumption about demonstrators who tend to execute outlier actions (Murphy, 2013). In such a case, the squared error in $\mathcal{H}$ is replaced by the absolute error (see Appendix A.3).

It should be mentioned that $q_{\boldsymbol{\psi}}(\mathbf{a}_t|\mathbf{s}_t, \mathbf{u}_t, k)$ maximizes the immediate reward and minimizes the weighted squared error between $\mathbf{u}_t$ and $\mathbf{a}_t$. The trade-off between the reward and squared-error is determined by $\mathbf{C}_{\boldsymbol{\omega}}(k)$. Specifically, for demonstrators with a small $\mathbf{C}_{\boldsymbol{\omega}}(k)$ (i.e., high-expertise demonstrators), the squared error has a large magnitude and $q_{\boldsymbol{\psi}}$ tends to minimize the squared error. Meanwhile, for demonstrators with a large value of $\mathbf{C}_{\boldsymbol{\omega}}(k)$ (i.e., low-expertise demonstrators), the squared error has a small magnitude and $q_{\boldsymbol{\psi}}$ tends to maximize the immediate reward.

We implement VILD with deep neural networks where we iteratively update $\boldsymbol{\phi}$, $\boldsymbol{\omega}$, and $\boldsymbol{\psi}$ by stochastic gradient methods, and update $\boldsymbol{\theta}$ by policy gradient methods. A pseudo-code of VILD and implementation details are given in Appendix B. In our implementation, we include a regularization term $L(\boldsymbol{\omega}) = T\mathbb{E}_{p(k)}[\log|\mathbf{C}_{\boldsymbol{\omega}}^{-1}(k)|]/2$, to penalize large value of $\mathbf{C}_{\boldsymbol{\omega}}(k)$. Without this regularization, $\mathbf{C}_{\boldsymbol{\omega}}(k)$ can be overly large which makes learning degenerate. We note that $\mathcal{H}$ already includes such a penalty via the trace term: $\mathbb{E}_{p(k)}[\text{Tr}(\mathbf{C}_{\boldsymbol{\omega}}^{-1}(k)\boldsymbol{\Sigma})]$. However, the strength of this penalty tends to be too small, since we choose $\boldsymbol{\Sigma}$ to be small.

VILD requires variable $k$ to be given along with demonstrations. However, There is no need for this variable to be provided by experts. When $k$ is not given, a simple strategy is to set $k = n$ and $K = N$. In other words, this strategy assumes that there is a one-to-one mapping between demonstration and demonstrator. We apply this strategy in our experiments with real-world demonstrations.

### 3.4 IMPORTANCE SAMPLING FOR REWARD LEARNING

To improve the convergence rate of VILD when updating $\boldsymbol{\phi}$, we use importance sampling (IS). Specifically, by analyzing the gradient $\nabla_{\boldsymbol{\phi}}\mathcal{H} = \nabla_{\boldsymbol{\phi}}\{\mathbb{E}_{p_{\text{d}}(\mathbf{s}_{1:T}, \mathbf{u}_{1:T}|k)p(k)}[\Sigma_{t=1}^{T}\mathbb{E}_{q_{\boldsymbol{\psi}}(\mathbf{a}_t|\mathbf{s}_t, \mathbf{u}_t, k)}[r_{\boldsymbol{\phi}}(\mathbf{s}_t, \mathbf{a}_t)]] - \mathbb{E}_{\tilde{q}_{\boldsymbol{\theta}}(\mathbf{s}_{1:T}, \mathbf{a}_{1:T})}[\Sigma_{t=1}^{T}r_{\boldsymbol{\phi}}(\mathbf{s}_t, \mathbf{a}_t)]\}$, we can see that the reward function is updated to maximize the expected cumulative reward obtained by demonstrators and $q_{\boldsymbol{\psi}}$, while minimizing the expected cumulative reward obtained by $q_{\boldsymbol{\theta}}$. However, low-quality demonstrations often yield low reward values. For this reason, stochastic gradients estimated by these demonstrations tend to be uninformative, which leads to slow convergence and poor data-efficiency.

To avoid estimating such uninformative gradients, we use IS to estimate gradients using high-quality demonstrations which are sampled with high probability. Briefly, IS is a technique for estimating an expectation over a distribution by using samples from a different distribution (Robert & Casella, 2005). For VILD, we propose to sample $k$ from a distribution $\tilde{p}(k) \propto \|\text{vec}(\mathbf{C}_{\boldsymbol{\omega}}^{-1}(k))\|_1$. This distribution assigns high probabilities to demonstrators with high estimated level of expertise (i.e., demonstrators with a small $\mathbf{C}_{\boldsymbol{\omega}}(k)$). With this distribution, the estimated gradients tend to be more informative which leads to a faster convergence. To reduce a sampling bias, we use a truncated importance weight: $w(k) = \min(p(k)/\tilde{p}(k), 1)$ (Ionides, 2008), which leads to an IS gradient: $\nabla_{\boldsymbol{\phi}}\mathcal{H}_{\text{IS}} = \nabla_{\boldsymbol{\phi}}\{\mathbb{E}_{p_{\text{d}}(\mathbf{s}_{1:T}, \mathbf{u}_{1:T}|k)\tilde{p}(k)}[w(k)\Sigma_{t=1}^{T}\mathbb{E}_{q_{\boldsymbol{\psi}}(\mathbf{a}_t|\mathbf{s}_t, \mathbf{u}_t, k)}[r_{\boldsymbol{\phi}}(\mathbf{s}_t, \mathbf{a}_t)]] - \mathbb{E}_{\tilde{q}_{\boldsymbol{\theta}}(\mathbf{s}_{1:T}, \mathbf{a}_{1:T})}[\Sigma_{t=1}^{T}r_{\boldsymbol{\phi}}(\mathbf{s}_t, \mathbf{a}_t)]\}$. Computing $w(k)$ requires $p(k)$, which can be estimated accurately since $k$ is a discrete random variable. For simplicity, we assume that $p(k)$ is a uniform distribution.

## 4 RELATED WORK

In this section, we will discuss a related area of supervised learning with diverse-quality data. Besides, we will discuss existing IL methods that use the variational approach.

**Supervised learning with diverse-quality data.**    In supervised learning, diverse-quality data has been extensively studied, e.g. learning with noisy labels (Angluin & Laird, 1988). This task assumes that human labelers may assign incorrect labels to training inputs. With such labelers, the obtained dataset consists of high-quality data with correct labels and low-quality data with incorrect labels. To handle this setting, many methods were proposed (Natarajan et al., 2013; Han et al., 2018). The most related methods are probabilistic models, which aim to infer correct labels and the level of labelers' expertise (Raykar et al., 2010; Khetan et al., 2018). Specifically, Raykar et al. (2010) proposed a method based on a two-coin model which enables estimating the correct labels and level of expertise. Recently, Khetan et al. (2018) proposed a method based on weighted loss functions, where the weight is determined by the estimated labels and level of expertise. Methods for supervised learning with diverse-quality data can be leveraged to learn a policy in our setting. However, they tend to perform poorly due to the issue of compounding error, as discussed previously in Section 3.1.

**Variational approach in IL.**    The variational approach has been previously utilized in IL to perform MM-IL and reduce over-fitting. Specifically, MM-IL aims to learn a multi-modal policy from diverse demonstrations collected by many experts (Li et al., 2017), where each mode of the policy represents decision making of each expert[2]. A multi-modal policy is commonly represented by a context-dependent policy, where each context represents each mode of the policy. The variational approach has been used to learn such contexts, i.e., by learning a variational auto-encoder (Wang et al., 2017) and maximizing a variational lower-bound of mutual information (Li et al., 2017; Hausman et al., 2017). Meanwhile, variational information bottleneck (VIB) (Alemi et al., 2017) has been used to reduce over-fitting in IL (Peng et al., 2019). Specifically, VIB aims to compress information flow by minimizing a variational bound of mutual information. This compression filters irrelevant signals, which leads to less over-fitting. Unlike these existing works, we utilize the variational approach to aid computing integrals in large state-action spaces, but not for learning a variational auto-encoder or optimizing a variational bound of mutual information.

## 5 EXPERIMENTS

In this section, we experimentally evaluate the performance of VILD (with and without IS) in continuous-control benchmarks and real-world crowdsourced demonstrations. For benchmarks, we use four continuous-control tasks from OpenAI gym (Brockman et al., 2016) with demonstrations from a pre-trained RL agent. For real-world demonstrations, we use a robosuite reaching task (Fan et al., 2018) with demonstrations from real-world crowdsourcing platform (Mandlekar et al., 2018). Performance is evaluated using a cumulative ground-truth reward along trajectories (i.e., higher is better) (Ho & Ermon, 2016), and this cumulative reward is computed using test trajectories generated by learned policies (i.e., $q_{\boldsymbol{\theta}}(\mathbf{a}_t|\mathbf{s}_t)$). We use 10 test trajectories for the benchmark tasks, and use 100 test trajectories for the robosuite reaching task. Note that we use a larger number of test trajectories due to high variability of initial states in the robosuite reaching task. We repeat experiments for 5 trials with different random seeds and report the mean and standard error.

**Baseline.**    We compare VILD against GAIL (Ho & Ermon, 2016), AIRL (Fu et al., 2018), VAIL (Peng et al., 2019), MaxEnt-IRL (Ziebart et al., 2010), and InfoGAIL (Li et al., 2017). These are online IL methods which collect transition samples to learn policies. We use trust region policy optimization (TRPO) (Schulman et al., 2015) to update policies, except for the Humanoid task where we use soft actor-critic (SAC) (Haarnoja et al., 2018). For InfoGAIL, we report the performance averaged over uniformly sampled contexts, as well as the performance with the best context chosen during testing.

---

[2]We emphasize that diverse demonstrations are different from diverse-quality demonstrations. Diverse demonstrations are collected by experts who execute equally good policies, while diverse-quality demonstrations are collected by mixed demonstrators; The former consists of demonstrations that are equally high-quality but diverse in behavior, while the latter consists of demonstrations that are diverse in both quality and behavior.

## 5.1 CONTINUOUS-CONTROL BENCHMARK TASKS

**Data generation.** To generate demonstrations from $\pi^\star$ (pre-trained by TRPO) according to Figure 1(b), we use two types of noisy policy $p(\mathbf{u}_t|\mathbf{a}_t, \mathbf{s}_t, k)$: Gaussian noisy policy: $\mathcal{N}(\mathbf{u}_t|\mathbf{a}_t, \sigma_k^2 \boldsymbol{I})$ and time-signal-dependent (TSD) noisy policy: $\mathcal{N}(\mathbf{u}_t|\mathbf{a}_t, \mathrm{diag}(\mathbf{b}_k(t) \times \|\mathbf{a}_t\|_1/d_\mathbf{a}))$, where $\mathbf{b}_k(t)$ is sampled from a noise process. We use $K = 10$ demonstrators with different $\sigma_k$ and noise processes for $\mathbf{b}_k(t)$. Each demonstrator generates trajectories with approximately $T = 1000$ time steps. The number of state-action pairs in each dataset is approximately 10000. Notice that for TSD, the noise variance depends on time and magnitude of actions. This characteristic of TSD has been observed in human motor control (van Beers et al., 2004). More details of data generation are given in Appendix C.

**Results against online IL methods.** Figure 2 shows learning curves of VILD and existing methods against the number of transition samples in HalfCheetah and Ant[3], whereas Table 1 reports the performance achieved in the last 100 iterations. Clearly, VILD with IS overall outperforms existing methods in terms of both data-efficiency and final performance, i.e., VILD with IS learns better policies using less numbers of transition samples. VILD without IS tends to outperform existing methods in terms of the final performance. However, it is less data-efficient when compared to VILD with IS, except on Humanoid with the Gaussian noisy policy, where VILD without IS tends to perform better than VILD with IS. We conjecture that this is because IS slightly biases gradient estimation, which may have a negative effect on the performance. Nonetheless, the overall good performance of VILD with IS suggests that it is an effective method to handle diverse-quality demonstrations.

On the contrary, existing methods perform poorly as expected, except on the Humanoid task. For the Humanoid task, VILD tends to perform the best in terms of the mean performance. Nonetheless, all methods except GAIL achieve statistically comparable performance according to t-test. This is perhaps because amateurs in this task perform relatively well compared to amateurs in other tasks, as seen from demonstrators' performance given in Table 2 and 3 (Appendix C). Since amateurs perform relatively well, demonstrations from these amateurs do not severely affect the performance of IL methods in this task when compared to the other tasks.

We found that InfoGAIL, which learns a context-dependent policy, may achieve good performance when the policy is conditioned on specific contexts. For instance, InfoGAIL (best context) performs quite well in the Walker2d task with the TSD noisy policy (the learning curves are provided in Figure 7(b)). However, as shown in Figure 10, its performance varies across contexts and is quite poor on average when using contexts from a uniform distribution. These results support our conjecture that MM-IL methods are not suitable for our setting where the level of demonstrators' expertise is absent.

It can be seen that VILD without IS performs better for the Gaussian noisy policy when compared to the TSD noisy policy. This is because the model of VILD is correctly specified for the Gaussian noisy policy, but the model is incorrectly specified for the TSD noisy policy; misspecified model indeed leads to the reduction in performance. Nonetheless, VILD with IS still performs well for both types of noisy policy. This is perhaps because negative effects of a misspecified model are not too severe for learning expertise parameters, which are required to compute $\widetilde{p}(k)$.

We also conduct the following evaluations. Due to space limitation, figures are given in Appendix D.

**Results against offline IL methods.** We compare VILD against offline IL methods based on supervised learning, namely behavior cloning (BC) (Pomerleau, 1988), Co-Teaching which is based on a method for learning with noisy labels (Han et al., 2018), and BC from diverse-quality demonstrations (BC-D) which optimizes the naive model described in Section 3.1. Results in Figure 8 show that these methods perform worse than VILD overall; BC performs the worst since it severely suffers from both the compounding error and low-quality demonstrations. Compared to BC, BC-D and Co-teaching are quite robust against low-quality demonstrations, but they still perform worse than VILD with IS.

**Accuracy of estimated expertise parameter.** To evaluate accuracy of estimated expertise parameter, we compare the ground-truth value of $\sigma_k$ under the Gaussian noisy policy against the learned covariance $\mathbf{C}_\omega(k)$. Figure 9 shows that VILD learns an accurate ranking of demonstrators' expertise. The values of these parameters are also quite accurate compared to the ground-truth, except for demonstrators with low-level of expertise. A reason for this phenomena is that low-quality demonstrations are highly dissimilar, which makes learning the expertise more challenging.

---

[3]Learning curves of other tasks are given in Figure 7 in Appendix D.

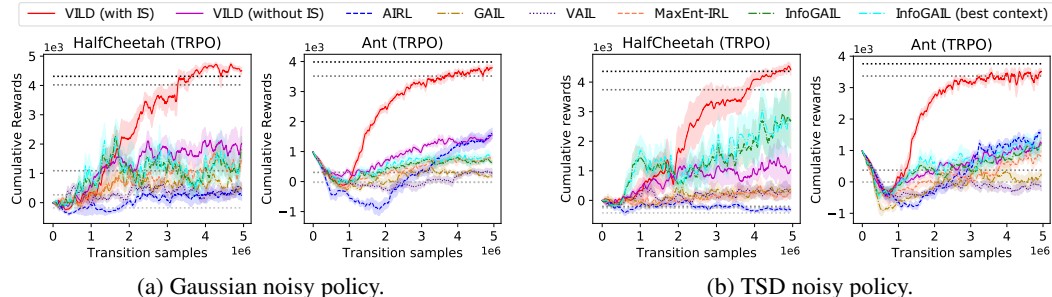

(a) Gaussian noisy policy.                          (b) TSD noisy policy.

Figure 2: Performance averaged over 5 trials in terms of the mean and standard error. Demonstrations are generated by 10 demonstrators using (a) Gaussian and (b) TSD noisy policies. Horizontal dotted lines indicate performance of $k = 1, 3, 5, 7, 10$ demonstrators. IS denotes importance sampling.

Table 1: Performance in the last 100 iterations in terms of the mean and standard error of cumulative rewards over 5 trials (higher is better). Boldfaces indicate best and comparable methods according to t-test with significance level 0.01. (G) denotes Gaussian noisy policy and (TSD) denotes time-signal-dependent noisy policy. The performance of VAIL is similar to that of GAIL and is omitted. The performance of InfoGAIL (best context) is overall similar to that of InfoGAIL and is also omitted.

| Task | VILD (IS) | VILD (w/o IS) | AIRL | GAIL | MaxEnt-IRL | InfoGAIL |
|---|---|---|---|---|---|---|
| HalfCheetah (G) | **4559 (43)** | 1848 (429) | 341 (177) | 551 (23) | 1192 (245) | 1244 (210) |
| HalfCheetah (TSD) | **4394 (136)** | 1159 (594) | -304 (51) | 318 (134) | 177 (132) | **2664 (779)** |
| Ant (G) | **3719 (65)** | 1426 (81) | 1417 (184) | 209 (30) | 731 (93) | 675 (36) |
| Ant (TSD) | **3396 (64)** | 1072 (134) | 1357 (59) | 97 (161) | 775 (135) | 1076 (140) |
| Walker2d (G) | **3470 (300)** | 2132 (64) | 1534 (99) | 1410 (115) | 1795 (172) | 1668 (82) |
| Walker2d (TSD) | **3115 (130)** | 1244 (132) | 578 (47) | 834 (84) | 752 (112) | 1041 (36) |
| Humanoid (G) | 3781 (557) | **4840 (56)** | 4274 (93) | 284 (24) | **3038 (731)** | **4047 (653)** |
| Humanoid (TSD) | **4600 (97)** | 3610 (448) | **4212 (121)** | 203 (31) | **4132 (651)** | **3962 (635)** |

## 5.2 ROBOSUITE REACHING TASK

In this experiment, we evaluate the robustness of VILD against real-world demonstrations. Specifically, we conduct an experiment using real-world demonstrations collected by a robotic crowdsourcing platform (Mandlekar et al., 2018). The public datasets were collected in the robosuite environment for object-manipulation tasks such as assembly tasks (Fan et al., 2018). In our experiment, we consider a reaching task, where demonstrations come from clipped assembly tasks when the robot's end-effector contacts the target object. We uses $N = 10$ demonstrations whose length are approximately $T = 500$ and set $K = 10$. The number of state-action pairs in a demonstration dataset is approximately 5000. For VILD, we apply the log-sigmoid function to the reward function, which improves the performance in this task. More details of the experimental setting are provided in Appendix C.2.

Figure 3 shows the performance of all methods, except VILD without IS and VAIL. We do not evaluate VILD without IS and VAIL since IS improves the performance and VAIL is comparable to GAIL. It can be seen that VILD with IS performs better than GAIL, AIRL, and MaxEnt-IRL. VILD also performs better than InfoGAIL in terms of the final performance; InfoGAIL learns faster in the early stage of learning, but its performance saturates and VILD eventually outperforms InfoGAIL. These experimental results show that VILD is more robust against real-world demonstrations with diverse-quality when compared to existing state-of-the-art methods. An example of trajectory generated by VILD's policy is shown in Figure 5.

Figure 4 shows the performance of InfoGAIL with different context variables $z$ (Li et al., 2017). We can see that InfoGAIL performs well when the policy is conditioned on specific contexts, e.g., $z = 7$. Indeed, the best context during testing can improve the performance of InfoGAIL. The effectiveness of such an approach is demonstrated in Figure 3, where InfoGAIL (best context) performs very well. However, InfoGAIL (best context) is less practical than VILD, since choosing the best context requires an expert to evaluate the performance of all contexts. In contrast, the performance of VILD does not depend on contexts, since VILD does not learn a context-dependent policy. Moreover, the

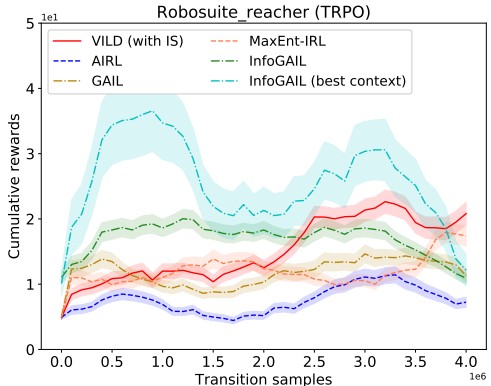

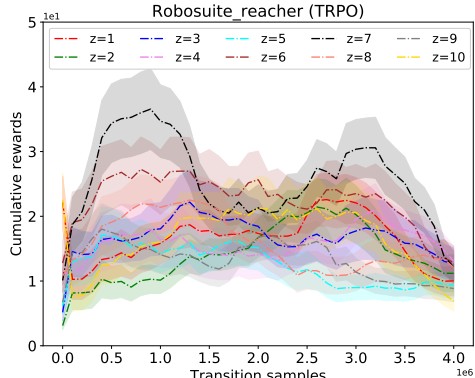

Figure 3: Performance of VILD with IS and baseline methods for the robosuite reaching task.

Figure 4: Performance of InfoGAIL with different contexts $z$ for the robosuite reaching task.

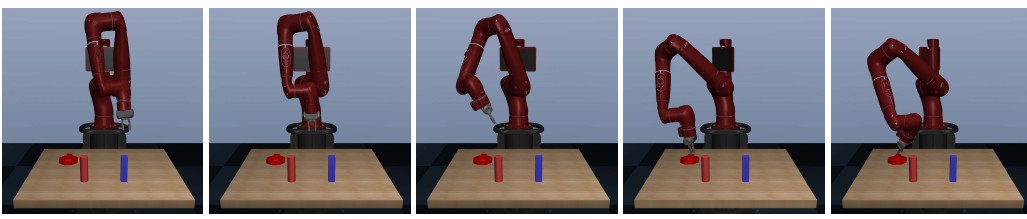

Figure 5: An example of trajectory generated by VILD's policy in the robosuite reaching task. The goal of the agent is to control the robot's end-effector to reach the red object. The value of reward function (for performance evaluation) is inverse proportion to the distance between the end-effector and the red object.

performance of InfoGAIL (best context) is quite unstable, and it is still outperformed by VILD in terms of the final performance.

## 6    CONCLUSION AND FUTURE WORK

In this paper, we explored a practical setting in IL where demonstrations have diverse-quality. We showed the deficiency of existing methods, and proposed a robust method called VILD, which learns both the reward function and the level of demonstrators' expertise by using the variational approach. Empirical results demonstrated that our work enables scalable and data-efficient IL under this practical setting.

In future, we will explore other approaches to efficiently estimate parameters of the proposed model except the variational approach. We will also explore approaches to handle model misspecification, i.e., scenarios where the noisy policy differs from the model $p_\omega$. Specifically, we will explore more flexible models of $p_\omega$ such as neural networks, as well as using the tempered posterior approach (Grünwald & van Ommen, 2017) to improve robustness of our model.

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

# A  DERIVATIONS

This section derives the lower-bounds of $f(\phi, \omega)$ and $g(\phi, \omega)$ presented in the paper. We also derive the objective function $\mathcal{H}(\phi, \omega, \psi, \theta)$ of VILD.

## A.1  LOWER-BOUND OF $f$

Let $l_{\phi, \omega}(\mathbf{s}_t, \mathbf{a}_t, \mathbf{u}_t, k)$ $=$ $r_\phi(\mathbf{s}_t, \mathbf{a}_t)$ $+$ $\log p_\omega(\mathbf{u}_t | \mathbf{s}_t, \mathbf{a}_t, k)$, we have that $f(\phi, \omega)$ $=$ $\mathbb{E}_{p_d(\mathbf{s}_{1:T}, \mathbf{u}_{1:T} | k) p(k)} \left[ \sum_{t=1}^{T} f_t(\phi, \omega) \right]$, where $f_t(\phi, \omega)$ $=$ $\log \int_{\mathcal{A}} \exp(l_{\phi, \omega}(\mathbf{s}_t, \mathbf{a}_t, \mathbf{u}_t, k)) \, \mathrm{d}\mathbf{a}_t$. By using a variational distribution $q_\psi(\mathbf{a}_t | \mathbf{s}_t, \mathbf{u}_t, k)$ with parameter $\psi$, we can bound $f_t(\phi, \omega)$ from below by using the Jensen inequality as follows:

$$
\begin{aligned}
f_t(\phi, \omega) &= \log \left( \int_{\mathcal{A}} \exp(l_{\phi, \omega}(\mathbf{s}_t, \mathbf{a}_t, \mathbf{u}_t, k)) \frac{q_\psi(\mathbf{a}_t | \mathbf{s}_t, \mathbf{u}_t, k)}{q_\psi(\mathbf{a}_t | \mathbf{s}_t, \mathbf{u}_t, k)} \mathrm{d}\mathbf{a}_t \right) \\
&\geqslant \int_{\mathcal{A}} q_\psi(\mathbf{a}_t | \mathbf{s}_t, \mathbf{u}_t, k) \log \left( \exp(l_{\phi, \omega}(\mathbf{s}_t, \mathbf{a}_t, \mathbf{u}_t, k)) \frac{1}{q_\psi(\mathbf{a}_t | \mathbf{s}_t, \mathbf{u}_t, k)} \right) \mathrm{d}\mathbf{a}_t \\
&= \mathbb{E}_{q_\psi(\mathbf{a}_t | \mathbf{s}_t, \mathbf{u}_t, k)} \left[ l_{\phi, \omega}(\mathbf{s}_t, \mathbf{a}_t, \mathbf{u}_t, k) - \log q_\psi(\mathbf{a}_t | \mathbf{s}_t, \mathbf{u}_t, k) \right] \\
&= \mathcal{F}_t(\phi, \omega, \psi).
\end{aligned}
\tag{6}
$$

Then, by using the linearity of expectation, we obtain the lower-bound of $f(\phi, \omega)$ as follows:

$$
\begin{aligned}
f(\phi, \omega) &\geqslant \mathbb{E}_{p_d(\mathbf{s}_{1:T}, \mathbf{u}_{1:T} | k) p(k)} \left[ \sum_{t=1}^{T} \mathcal{F}_t(\phi, \omega, \psi) \right] \\
&= \mathbb{E}_{p_d(\mathbf{s}_{1:T}, \mathbf{u}_{1:T} | k) p(k)} \left[ \sum_{t=1}^{T} \mathbb{E}_{q_\psi(\mathbf{a}_t | \mathbf{s}_t, \mathbf{u}_t, k)} \left[ l_{\phi, \omega}(\mathbf{s}_t, \mathbf{a}_t, \mathbf{u}_t, k) - \log q_\psi(\mathbf{a}_t | \mathbf{s}_t, \mathbf{u}_t, k) \right] \right] \\
&= \mathcal{F}(\phi, \omega, \psi).
\end{aligned}
\tag{7}
$$

To verify that $f(\phi, \omega) = \max_\psi \mathcal{F}(\phi, \omega, \psi)$, we maximize $\mathcal{F}_t(\phi, \omega, \psi)$ w.r.t. $q_\psi$ under the constraint that $q_\psi$ is a valid probability density, i.e., $q_\psi(\mathbf{a}_t | \mathbf{s}_t, \mathbf{u}_t, k) > 0$ and $\int_{\mathcal{A}} q_\psi(\mathbf{a}_t | \mathbf{s}_t, \mathbf{u}_t, k) \mathrm{d}\mathbf{a}_t = 1$. By setting the derivative of $\mathcal{F}_t(\phi, \omega, \psi)$ w.r.t. $q_\psi$ to zero, we obtain

$$
\begin{aligned}
q_\psi(\mathbf{a}_t | \mathbf{s}_t, \mathbf{u}_t, k) &= \exp(l_{\phi, \omega}(\mathbf{s}_t, \mathbf{a}_t, \mathbf{u}_t, k) - 1) \\
&= \frac{\exp(l_{\phi, \omega}(\mathbf{s}_t, \mathbf{a}_t, \mathbf{u}_t, k))}{\int_{\mathcal{A}} \exp(l_{\phi, \omega}(\mathbf{s}_t, \mathbf{a}_t, \mathbf{u}_t, k)) \, \mathrm{d}\mathbf{a}_t},
\end{aligned}
$$

where the last line follows from the constraint $\int_{\mathcal{A}} q_\psi(\mathbf{a}_t | \mathbf{s}_t, \mathbf{u}_t, k) \mathrm{d}\mathbf{a}_t = 1$. To show that this is indeed the maximizer, we substitute $q_{\psi^\star}(\mathbf{a}_t | \mathbf{s}_t, \mathbf{u}_t, k) = \frac{\exp(l(\mathbf{s}_t, \mathbf{a}_t, \mathbf{u}_t, k))}{\int_{\mathcal{A}} \exp(l(\mathbf{s}_t, \mathbf{a}_t, \mathbf{u}_t, k)) \mathrm{d}\mathbf{a}_t}$ into $\mathcal{F}_t(\phi, \omega, \psi)$:

$$
\begin{aligned}
\mathcal{F}_t(\phi, \omega, \psi^\star) &= \mathbb{E}_{q_\psi^\star(\mathbf{a}_t | \mathbf{s}_t, \mathbf{u}_t, k)} \left[ l_{\phi, \omega}(\mathbf{s}_t, \mathbf{a}_t, \mathbf{u}_t, k) - \log q_{\psi^\star}(\mathbf{a}_t | \mathbf{s}_t, \mathbf{u}_t, k) \right] \\
&= \log \left( \int_{\mathcal{A}} \exp(l_{\phi, \omega}(\mathbf{s}_t, \mathbf{a}_t, \mathbf{u}_t, k)) \, \mathrm{d}\mathbf{a}_t \right).
\end{aligned}
$$

This equality verifies that $f_t(\phi, \omega) = \max_\psi \mathcal{F}_t(\phi, \omega, \psi)$. Finally, by using the linearity of expectation, we have that $f(\phi, \omega) = \max_\psi \mathcal{F}(\phi, \omega, \psi)$.

## A.2  LOWER-BOUND OF $g$

Next, we derive the lower-bound of $g(\phi, \omega)$ presented in the paper. We first derive a trivial lower-bound using a general variational distribution over trajectories and reveal its issues. Then, we derive a lower-bound presented in the paper by using a structured variational distribution. Recall that the function $g(\phi, \omega) = \log Z_{\phi, \omega}$ is

$$
g(\phi, \omega) = \log \left( \sum_{k=1}^{K} p(k) \int \cdots \int_{(\mathcal{S} \times \mathcal{A} \times \mathcal{A})^T} p_1(\mathbf{s}_1) \prod_{t=1}^{T} p(\mathbf{s}_{t+1} | \mathbf{s}_t, \mathbf{u}_t) \exp(l(\mathbf{s}_t, \mathbf{a}_t, \mathbf{u}_t, k)) \, \mathrm{d}\mathbf{s}_{1:T} \mathrm{d}\mathbf{u}_{1:T} \mathrm{d}\mathbf{a}_{1:T} \right).
$$

**Lower-bound via a variational distribution**  A lower-bound of $g$ can be obtained by using a variational distribution $\bar{q}_\beta(\mathbf{s}_{1:T}, \mathbf{u}_{1:T}, \mathbf{a}_{1:T}, k)$ with parameter $\beta$. We note that this variational distribution allows any dependency between the random variables $\mathbf{s}_{1:T}$, $\mathbf{u}_{1:T}$, $\mathbf{a}_{1:T}$, and $k$. By using this distribution, we have a

lower-bound

$$
\begin{aligned}
g(\boldsymbol{\phi}, \boldsymbol{\omega}) = \log \Bigg( & \sum_{k=1}^{K} p(k) \int \cdots \int_{(\mathcal{S} \times \mathcal{A} \times \mathcal{A})^T} p_1(\mathbf{s}_1) \prod_{t=1}^{T} p(\mathbf{s}_{t+1} | \mathbf{s}_t, \mathbf{u}_t) \exp\left(l_{\boldsymbol{\phi}, \boldsymbol{\omega}}(\mathbf{s}_t, \mathbf{a}_t, \mathbf{u}_t, k)\right) \\
& \times \frac{\bar{q}_{\boldsymbol{\beta}}(\mathbf{s}_{1:T}, \mathbf{u}_{1:T}, \mathbf{a}_{1:T}, k)}{\bar{q}_{\boldsymbol{\beta}}(\mathbf{s}_{1:T}, \mathbf{u}_{1:T}, \mathbf{a}_{1:T}, k)} \mathrm{d}\mathbf{s}_{1:T} \mathrm{d}\mathbf{u}_{1:T} \mathrm{d}\mathbf{a}_{1:T} \Bigg) \\
\geqslant\ & \mathbb{E}_{\bar{q}_{\boldsymbol{\beta}}(\mathbf{s}_{1:T}, \mathbf{u}_{1:T}, \mathbf{a}_{1:T}, k)} \Bigg[ \log p(k) p_1(\mathbf{s}_1) + \sum_{t=1}^{T} \{ \log p(\mathbf{s}_{t+1} | \mathbf{s}_t, \mathbf{u}_t) + l_{\boldsymbol{\phi}, \boldsymbol{\omega}}(\mathbf{s}_t, \mathbf{a}_t, \mathbf{u}_t, k) \} \\
& \quad - \log \bar{q}_{\boldsymbol{\beta}}(\mathbf{s}_{1:T}, \mathbf{u}_{1:T}, \mathbf{a}_{1:T}, k) \Bigg] \\
=\ & \bar{\mathcal{G}}(\boldsymbol{\phi}, \boldsymbol{\omega}, \boldsymbol{\beta}).
\end{aligned}
\tag{8}
$$

The main issue of using this lower-bound is that, $\bar{\mathcal{G}}(\boldsymbol{\phi}, \boldsymbol{\omega}, \boldsymbol{\beta})$ can be computed or approximated only when we have an access to the transition probability $p(\mathbf{s}_{t+1} | \mathbf{s}_t, \mathbf{u}_t)$. In many practical tasks, the transition probability is unknown and needs to be estimated. However, estimating the transition probability for large state and action spaces is known to be highly challenging (Sutton & Barto, 1998). For these reasons, this lower-bound is not suitable for our method.

**Lower-bound via a structured variational distribution**  To avoid the above issue, we use the structure variational approach (Hoffman & Blei, 2015), where the key idea is to pre-define conditional dependency to ease computation. Specifically, we use a variational distribution $q_{\boldsymbol{\theta}}(\mathbf{a}_t, \mathbf{u}_t | \mathbf{s}_t, k)$ with parameter $\boldsymbol{\theta}$ and define dependencies between states according to the transition probability of an MDP. With this variational distribution, we lower-bound $g$ as follows:

$$
\begin{aligned}
g(\boldsymbol{\phi}, \boldsymbol{\omega}) = \log \Bigg( & \sum_{k=1}^{K} p(k) \int \cdots \int_{(\mathcal{S} \times \mathcal{A} \times \mathcal{A})^T} p_1(\mathbf{s}_1) \prod_{t=1}^{T} p(\mathbf{s}_{t+1} | \mathbf{s}_t, \mathbf{u}_t) \exp\left(l_{\boldsymbol{\phi}, \boldsymbol{\omega}}(\mathbf{s}_t, \mathbf{a}_t, \mathbf{u}_t, k)\right) \\
& \times \frac{q_{\boldsymbol{\theta}}(\mathbf{a}_t, \mathbf{u}_t | \mathbf{s}_t, k)}{q_{\boldsymbol{\theta}}(\mathbf{a}_t, \mathbf{u}_t | \mathbf{s}_t, k)} \mathrm{d}\mathbf{s}_{1:T} \mathrm{d}\mathbf{u}_{1:T} \mathrm{d}\mathbf{a}_{1:T} \Bigg) \\
\geqslant\ & \mathbb{E}_{\tilde{q}_{\boldsymbol{\theta}}(\mathbf{s}_{1:T}, \mathbf{u}_{1:T}, \mathbf{a}_{1:T}, k)} \Bigg[ \sum_{t=1}^{T} l_{\boldsymbol{\phi}, \boldsymbol{\omega}}(\mathbf{s}_t, \mathbf{a}_t, \mathbf{u}_t, k) - \log q_{\boldsymbol{\theta}}(\mathbf{a}_t, \mathbf{u}_t | \mathbf{s}_t, k) \Bigg] \\
=\ & \mathcal{G}(\boldsymbol{\phi}, \boldsymbol{\omega}, \boldsymbol{\theta}),
\end{aligned}
\tag{9}
$$

where $\tilde{q}_{\boldsymbol{\theta}}(\mathbf{s}_{1:T}, \mathbf{u}_{1:T}, \mathbf{a}_{1:T}, k) = p(k) p_1(\mathbf{s}_1) \Pi_{t=1}^{T} p(\mathbf{s}_{t+1} | \mathbf{s}_t, \mathbf{u}_t) q_{\boldsymbol{\theta}}(\mathbf{a}_t, \mathbf{u}_t | \mathbf{s}_t, k)$. The optimal variational distribution $q_{\boldsymbol{\theta}^\star}(\mathbf{a}_t, \mathbf{u}_t | \mathbf{s}_t, k)$ can be founded by maximizing $\mathcal{G}(\boldsymbol{\phi}, \boldsymbol{\omega}, \boldsymbol{\theta})$ w.r.t. $q_{\boldsymbol{\theta}}$. Solving this maximization problem is identical to solving a maximum entropy RL (MaxEnt-RL) problem (Ziebart et al., 2010) for an MDP defined by a tuple $\mathcal{M} = (\mathcal{S} \times \mathbb{N}_+, \mathcal{A} \times \mathcal{A}, p(\mathbf{s}', | \mathbf{s}, \mathbf{u}) \mathbb{I}_{k=k'}, p_1(\mathbf{s}_1) p(k_1), l_{\boldsymbol{\phi}, \boldsymbol{\omega}}(\mathbf{s}, \mathbf{a}, \mathbf{u}, k))$. Specifically, this MDP is defined with a state variable $(\mathbf{s}_t, k_t) \in \mathcal{S} \times \mathbb{N}$, an action variable $(\mathbf{a}_t, \mathbf{u}_t) \in \mathcal{A} \times \mathcal{A}$, a transition probability density $p(\mathbf{s}_{t+1}, | \mathbf{s}_t, \mathbf{u}_t) \mathbb{I}_{k_t = k_{t+1}}$, an initial state density $p_1(\mathbf{s}_1) p(k_1)$, and a reward function $l_{\boldsymbol{\phi}, \boldsymbol{\omega}}(\mathbf{s}, \mathbf{a}, \mathbf{u}, k)$. Here, $\mathbb{I}_{a=b}$ is the indicator function which equals to 1 if $a = b$ and 0 otherwise. By adopting the optimality results of MaxEnt-RL (Ziebart et al., 2010; Haarnoja et al., 2018), we have $g(\boldsymbol{\phi}, \boldsymbol{\omega}) = \max_{\boldsymbol{\theta}} \mathcal{G}(\boldsymbol{\phi}, \boldsymbol{\omega}, \boldsymbol{\theta})$, where the optimal variational distribution is

$$
q_{\boldsymbol{\theta}^\star}(\mathbf{a}_t, \mathbf{u}_t | \mathbf{s}_t, k) = \exp(Q(\mathbf{s}_t, k, \mathbf{a}_t, \mathbf{u}_t) - V(\mathbf{s}_t, k)).
\tag{10}
$$

The functions $Q$ and $V$ are soft-value functions defined as

$$
Q(\mathbf{s}_t, k, \mathbf{a}_t, \mathbf{u}_t) = l_{\boldsymbol{\phi}, \boldsymbol{\omega}}(\mathbf{s}_t, \mathbf{a}_t, \mathbf{u}_t, k) + \mathbb{E}_{p(\mathbf{s}_{t+1} | \mathbf{s}_t, \mathbf{u}_t)} \left[ V(\mathbf{s}_{t+1}, k) \right],
\tag{11}
$$

$$
V(\mathbf{s}_t, k) = \log \iint_{\mathcal{A} \times \mathcal{A}} \exp\left(Q(\mathbf{s}_t, k, \mathbf{a}_t, \mathbf{u}_t)\right) \mathrm{d}\mathbf{a}_t \mathrm{d}\mathbf{u}_t.
\tag{12}
$$

## A.3   OBJECTIVE FUNCTION $\mathcal{H}$ OF VILD

This section derives the objective function $\mathcal{H}(\boldsymbol{\phi}, \boldsymbol{\omega}, \boldsymbol{\psi}, \boldsymbol{\theta})$ from $\mathcal{F}(\boldsymbol{\phi}, \boldsymbol{\omega}, \boldsymbol{\psi}) - \mathcal{G}(\boldsymbol{\phi}, \boldsymbol{\omega}, \boldsymbol{\theta})$. Specifically, we substitute the models $p_{\boldsymbol{\omega}}(\mathbf{u}_t | \mathbf{s}_t, \mathbf{a}_t, k) = \mathcal{N}(\mathbf{u}_t | \mathbf{a}_t, \mathbf{C}_{\boldsymbol{\omega}}(k))$ and $q_{\boldsymbol{\theta}}(\mathbf{a}_t, \mathbf{u}_t | \mathbf{s}_t, k) = q_{\boldsymbol{\theta}}(\mathbf{a}_t | \mathbf{s}_t) \mathcal{N}(\mathbf{u}_t | \mathbf{a}_t, \boldsymbol{\Sigma})$. We also give an example when using a Laplace distribution for $p_{\boldsymbol{\omega}}(\mathbf{u}_t | \mathbf{s}_t, \mathbf{a}_t, k)$ instead of the Gaussian distribution.

First, we substitute $q_{\boldsymbol{\theta}}(\mathbf{a}_t, \mathbf{u}_t | \mathbf{s}_t, k) = q_{\boldsymbol{\theta}}(\mathbf{a}_t | \mathbf{s}_t) \mathcal{N}(\mathbf{u}_t | \mathbf{a}_t, \boldsymbol{\Sigma})$ into $\mathcal{G}$:

$$\mathcal{G}(\boldsymbol{\phi}, \boldsymbol{\omega}, \boldsymbol{\theta}) = \mathbb{E}_{\tilde{q}_{\boldsymbol{\theta}}(\mathbf{s}_{1:T}, \mathbf{u}_{1:T}, \mathbf{a}_{1:T}, k)} \left[ \sum_{t=1}^{T} l_{\boldsymbol{\phi}, \boldsymbol{\omega}}(\mathbf{s}_t, \mathbf{a}_t, \mathbf{u}_t, k) - \log \mathcal{N}(\mathbf{u}_t | \mathbf{a}_t, \boldsymbol{\Sigma}) - \log q_{\boldsymbol{\theta}}(\mathbf{a}_t | \mathbf{s}_t) \right]$$

$$= \mathbb{E}_{\tilde{q}_{\boldsymbol{\theta}}(\mathbf{s}_{1:T}, \mathbf{u}_{1:T}, \mathbf{a}_{1:T}, k)} \left[ \sum_{t=1}^{T} l_{\boldsymbol{\phi}, \boldsymbol{\omega}}(\mathbf{s}_t, \mathbf{a}_t, \mathbf{u}_t, k) + \frac{1}{2} \|\mathbf{u}_t - \mathbf{a}_t\|^2_{\boldsymbol{\Sigma}^{-1}} - \log q_{\boldsymbol{\theta}}(\mathbf{a}_t | \mathbf{s}_t) \right] + c_1,$$

where $c_1$ is a constant corresponding to the log-normalization term of the Gaussian distribution. Next, by using the re-parameterization trick, we rewrite $\tilde{q}_{\boldsymbol{\theta}}(\mathbf{s}_{1:T}, \mathbf{u}_{1:T}, \mathbf{a}_{1:T}, k)$ as

$$\tilde{q}_{\boldsymbol{\theta}}(\mathbf{s}_{1:T}, \mathbf{u}_{1:T}, \mathbf{a}_{1:T}, k) = p(k) p_1(\mathbf{s}_1) \prod_{t=1}^{T} p_1(\mathbf{s}_{t+1} | \mathbf{s}_t, \mathbf{a}_t + \boldsymbol{\Sigma}^{1/2} \boldsymbol{\epsilon}_t) \mathcal{N}(\boldsymbol{\epsilon}_t | 0, \boldsymbol{I}) q_{\boldsymbol{\theta}}(\mathbf{a}_t | \mathbf{s}_t),$$

where we use $\mathbf{u}_t = \mathbf{a}_t + \boldsymbol{\Sigma}^{1/2} \boldsymbol{\epsilon}_t$ with $\boldsymbol{\epsilon}_t \sim \mathcal{N}(\boldsymbol{\epsilon}_t | 0, \boldsymbol{I})$. With this, the expectation of $\Sigma_{t=1}^{T} \|\mathbf{u}_t - \mathbf{a}_t\|^2_{\boldsymbol{\Sigma}^{-1}}$ over $\tilde{q}_{\boldsymbol{\theta}}(\mathbf{s}_{1:T}, \mathbf{u}_{1:T}, \mathbf{a}_{1:T}, k)$ can be written as

$$\mathbb{E}_{\tilde{q}_{\boldsymbol{\theta}}(\mathbf{s}_{1:T}, \mathbf{u}_{1:T}, \mathbf{a}_{1:T}, k)} \left[ \sum_{t=1}^{T} \|\mathbf{u}_t - \mathbf{a}_t\|^2_{\boldsymbol{\Sigma}^{-1}} \right] = \mathbb{E}_{\tilde{q}_{\boldsymbol{\theta}}(\mathbf{s}_{1:T}, \mathbf{u}_{1:T}, \mathbf{a}_{1:T}, k)} \left[ \sum_{t=1}^{T} \|\mathbf{a}_t + \boldsymbol{\Sigma}^{1/2} \boldsymbol{\epsilon}_t - \mathbf{a}_t\|^2_{\boldsymbol{\Sigma}^{-1}} \right]$$

$$= \mathbb{E}_{\tilde{q}_{\boldsymbol{\theta}}(\mathbf{s}_{1:T}, \mathbf{u}_{1:T}, \mathbf{a}_{1:T}, k)} \left[ \sum_{t=1}^{T} \|\boldsymbol{\Sigma}^{1/2} \boldsymbol{\epsilon}_t\|^2_{\boldsymbol{\Sigma}^{-1}} \right]$$

$$= T d_{\mathbf{a}},$$

which is a constant. Then, the quantity $\mathcal{G}$ can be expressed as

$$\mathcal{G}(\boldsymbol{\phi}, \boldsymbol{\omega}, \boldsymbol{\theta}) = \mathbb{E}_{\tilde{q}_{\boldsymbol{\theta}}(\mathbf{s}_{1:T}, \mathbf{u}_{1:T}, \mathbf{a}_{1:T}, k)} \left[ \sum_{t=1}^{T} l_{\boldsymbol{\phi}, \boldsymbol{\omega}}(\mathbf{s}_t, \mathbf{a}_t, \mathbf{u}_t, k) - \log q_{\boldsymbol{\theta}}(\mathbf{a}_t | \mathbf{s}_t) \right] + c_1 + T d_{\mathbf{a}}.$$

By ignoring the constant, the optimization problem $\max_{\boldsymbol{\phi}, \boldsymbol{\omega}, \psi} \min_{\boldsymbol{\theta}} \mathcal{F}(\boldsymbol{\phi}, \boldsymbol{\omega}, \psi) - \mathcal{G}(\boldsymbol{\phi}, \boldsymbol{\omega}, \boldsymbol{\theta})$ is equivalent to

$$\max_{\boldsymbol{\phi}, \boldsymbol{\omega}, \psi} \min_{\boldsymbol{\theta}} \mathbb{E}_{p_{\mathrm{d}}(\mathbf{s}_{1:T}, \mathbf{u}_{1:T} | k) p(k)} \left[ \sum_{t=1}^{T} \mathbb{E}_{q_\psi(\mathbf{a}_t | \mathbf{s}_t, \mathbf{u}_t, k)} \left[ l_{\boldsymbol{\phi}, \boldsymbol{\omega}}(\mathbf{s}_t, \mathbf{a}_t, \mathbf{u}_t, k) - \log q_\psi(\mathbf{a}_t | \mathbf{s}_t, \mathbf{u}_t, k) \right] \right]$$

$$- \mathbb{E}_{\tilde{q}_{\boldsymbol{\theta}}(\mathbf{s}_{1:T}, \mathbf{u}_{1:T}, \mathbf{a}_{1:T}, k)} \left[ \sum_{t=1}^{T} l_{\boldsymbol{\phi}, \boldsymbol{\omega}}(\mathbf{s}_t, \mathbf{a}_t, \mathbf{u}_t, k) - \log q_{\boldsymbol{\theta}}(\mathbf{a}_t | \mathbf{s}_t) \right]. \tag{13}$$

Our next step is to substitute $p_{\boldsymbol{\omega}}(\mathbf{u}_t | \mathbf{s}_t, \mathbf{a}_t, k)$ by our choice of model. First, let us consider a Gaussian distribution $p_{\boldsymbol{\omega}}(\mathbf{u}_t | \mathbf{s}_t, \mathbf{a}_t, k) = \mathcal{N}(\mathbf{u}_t | \mathbf{a}_t, \mathbf{C}_{\boldsymbol{\omega}}(\mathbf{s}_t, k))$, where the covariance depends on state. With this model, the second term in Eq. (13) is given by

$$\mathbb{E}_{\tilde{q}_{\boldsymbol{\theta}}(\mathbf{s}_{1:T}, \mathbf{u}_{1:T}, \mathbf{a}_{1:T}, k)} \left[ \sum_{t=1}^{T} l_{\boldsymbol{\phi}, \boldsymbol{\omega}}(\mathbf{s}_t, \mathbf{a}_t, \mathbf{u}_t, k) - \log q_{\boldsymbol{\theta}}(\mathbf{a}_t | \mathbf{s}_t) \right]$$

$$= \mathbb{E}_{\tilde{q}_{\boldsymbol{\theta}}(\mathbf{s}_{1:T}, \mathbf{u}_{1:T}, \mathbf{a}_{1:T}, k)} \left[ \sum_{t=1}^{T} r_{\boldsymbol{\phi}}(\mathbf{s}_t, \mathbf{a}_t) + \log \mathcal{N}(\mathbf{u}_t | \mathbf{a}_t, \mathbf{C}_{\boldsymbol{\omega}}(\mathbf{s}_t, k)) - \log q_{\boldsymbol{\theta}}(\mathbf{a}_t | \mathbf{s}_t) \right]$$

$$= \mathbb{E}_{\tilde{q}_{\boldsymbol{\theta}}(\mathbf{s}_{1:T}, \mathbf{u}_{1:T}, \mathbf{a}_{1:T}, k)} \left[ \sum_{t=1}^{T} r_{\boldsymbol{\phi}}(\mathbf{s}_t, \mathbf{a}_t) - \frac{1}{2} \|\mathbf{u}_t - \mathbf{a}_t\|^2_{\mathbf{C}_{\boldsymbol{\omega}}^{-1}(\mathbf{s}_t, k)} - \frac{1}{2} \log |\mathbf{C}_{\boldsymbol{\omega}}(\mathbf{s}_t, k)| - \log q_{\boldsymbol{\theta}}(\mathbf{a}_t | \mathbf{s}_t) \right] + c_2,$$

where $c_2 = -\frac{d_{\mathbf{a}}}{2} \log 2\pi$ is a constant. By using the reparameterization trick, we write the expectation of $\Sigma_{t=1}^{T} \|\mathbf{u}_t - \mathbf{a}_t\|^2_{\mathbf{C}_{\boldsymbol{\omega}}^{-1}(\mathbf{s}_t, k)}$ as follows:

$$\mathbb{E}_{\tilde{q}_{\boldsymbol{\theta}}(\mathbf{s}_{1:T}, \mathbf{u}_{1:T}, \mathbf{a}_{1:T}, k)} \left[ \sum_{t=1}^{T} \|\mathbf{u}_t - \mathbf{a}_t\|^2_{\mathbf{C}_{\boldsymbol{\omega}}^{-1}(\mathbf{s}_t, k)} \right] = \mathbb{E}_{\tilde{q}_{\boldsymbol{\theta}}(\mathbf{s}_{1:T}, \mathbf{u}_{1:T}, \mathbf{a}_{1:T}, k)} \left[ \sum_{t=1}^{T} \|\mathbf{a}_t + \boldsymbol{\Sigma}^{1/2} \boldsymbol{\epsilon}_t - \mathbf{a}_t\|^2_{\mathbf{C}_{\boldsymbol{\omega}}^{-1}(\mathbf{s}_t, k)} \right]$$

$$= \mathbb{E}_{\tilde{q}_{\boldsymbol{\theta}}(\mathbf{s}_{1:T}, \mathbf{u}_{1:T}, \mathbf{a}_{1:T}, k)} \left[ \sum_{t=1}^{T} \|\boldsymbol{\Sigma}^{1/2} \boldsymbol{\epsilon}_t\|^2_{\mathbf{C}_{\boldsymbol{\omega}}^{-1}(\mathbf{s}_t, k)} \right].$$

Using this equality, the second term in Eq. (13) is given by

$$\mathbb{E}_{\tilde{q}_{\boldsymbol{\theta}}(\mathbf{s}_{1:T}, \mathbf{u}_{1:T}, \mathbf{a}_{1:T}, k)} \left[ \sum_{t=1}^{T} r_{\boldsymbol{\phi}}(\mathbf{s}_t, \mathbf{a}_t) - \log q_{\boldsymbol{\theta}}(\mathbf{a}_t | \mathbf{s}_t) - \frac{1}{2} \left( \|\boldsymbol{\Sigma}^{1/2} \boldsymbol{\epsilon}_t\|^2_{\mathbf{C}_{\boldsymbol{\omega}}^{-1}(\mathbf{s}_t, k)} + \log |\mathbf{C}_{\boldsymbol{\omega}}(\mathbf{s}_t, k)| \right) \right]. \tag{14}$$

Maximizing this quantity w.r.t. $\boldsymbol{\theta}$ has an implication as follows: $q_{\boldsymbol{\theta}}(\mathbf{a}_t|\mathbf{s}_t)$ maximizes the expected cumulative reward while avoiding states that are difficult for demonstrators. Specifically, a large value of $\mathbb{E}_{p(k)}\left[\log|\mathbf{C}_{\boldsymbol{\omega}}(\mathbf{s}_t,k)|\right]$ indicates that demonstrators have a low level of expertise for state $\mathbf{s}_t$ on average, given by our estimated covariance. In other words, this state is difficult to accurately execute optimal actions for all demonstrators on averages. Since the policy $q_{\boldsymbol{\theta}}(\mathbf{a}_t|\mathbf{s}_t)$ should minimize $\mathbb{E}_{p(k)}\left[\log|\mathbf{C}_{\boldsymbol{\omega}}(\mathbf{s}_t,k)|\right]$, the policy should avoid states that are difficult for demonstrators. We expect that this property may improve exploration-exploitation trade-off in IL. Nonetheless, we leave an investigation of this property for future work, since this is not in the scope of the paper.

In this paper, we specify that the covariance does not depend on state: $\mathbf{C}_{\boldsymbol{\omega}}(\mathbf{s}_t,k) = \mathbf{C}_{\boldsymbol{\omega}}(k)$. This model specification enables us to simplify Eq. (14) as follows:

$$\mathbb{E}_{\widetilde{q}_{\boldsymbol{\theta}}(\mathbf{s}_{1:T},\mathbf{u}_{1:T},\mathbf{a}_{1:T},k)}\left[\sum_{t=1}^{T} r_{\boldsymbol{\phi}}(\mathbf{s}_t,\mathbf{a}_t) - \log q_{\boldsymbol{\theta}}(\mathbf{a}_t|\mathbf{s}_t) - \frac{1}{2}\left(\|\boldsymbol{\Sigma}^{1/2}\boldsymbol{\epsilon}_t\|_{\mathbf{C}_{\boldsymbol{\omega}}^{-1}(k)}^2 + \log|\mathbf{C}_{\boldsymbol{\omega}}(k)|\right)\right]$$

$$= \mathbb{E}_{\widetilde{q}_{\boldsymbol{\theta}}(\mathbf{s}_{1:T},\mathbf{u}_{1:T},\mathbf{a}_{1:T},k)}\left[\sum_{t=1}^{T} r_{\boldsymbol{\phi}}(\mathbf{s}_t,\mathbf{a}_t) - \log q_{\boldsymbol{\theta}}(\mathbf{a}_t|\mathbf{s}_t)\right] - \frac{T}{2}\mathbb{E}_{p(k)\mathcal{N}(\boldsymbol{\epsilon}|0,\boldsymbol{I})}\left[\|\boldsymbol{\Sigma}^{1/2}\boldsymbol{\epsilon}\|_{\mathbf{C}_{\boldsymbol{\omega}}^{-1}(k)}^2 + \log|\mathbf{C}_{\boldsymbol{\omega}}(k)|\right]$$

$$= \mathbb{E}_{\widetilde{q}_{\boldsymbol{\theta}}(\mathbf{s}_{1:T},\mathbf{a}_{1:T})}\left[\sum_{t=1}^{T} r_{\boldsymbol{\phi}}(\mathbf{s}_t,\mathbf{a}_t) - \log q_{\boldsymbol{\theta}}(\mathbf{a}_t|\mathbf{s}_t)\right] - \frac{T}{2}\mathbb{E}_{p(k)}\left[\mathrm{Tr}(\mathbf{C}_{\boldsymbol{\omega}}^{-1}(k)\boldsymbol{\Sigma}) + \log|\mathbf{C}_{\boldsymbol{\omega}}(k)|\right],$$

where $\widetilde{q}_{\boldsymbol{\theta}}(\mathbf{s}_{1:T},\mathbf{a}_{1:T}) = p_1(\mathbf{s}_1)\prod_{t=1}^{T}\int_{\mathcal{A}} p(\mathbf{s}_{t+1}|\mathbf{s}_t,\mathbf{u}_t)\mathcal{N}(\mathbf{u}_t|\mathbf{a}_t,\boldsymbol{\Sigma})\mathrm{d}\mathbf{u}_t q_{\boldsymbol{\theta}}(\mathbf{a}_t|\mathbf{s}_t)$. The last line follows from the quadratic form identity: $\mathbb{E}_{\mathcal{N}(\boldsymbol{\epsilon}_t|0,\boldsymbol{I})}\left[\|\boldsymbol{\Sigma}^{1/2}\boldsymbol{\epsilon}_t\|_{\mathbf{C}_{\boldsymbol{\omega}}^{-1}(k)}^2\right] = \mathrm{Tr}(\mathbf{C}_{\boldsymbol{\omega}}^{-1}(k)\boldsymbol{\Sigma})$. Next, we substitute $p_{\boldsymbol{\omega}}(\mathbf{u}_t|\mathbf{s}_t,\mathbf{a}_t,k) = \mathcal{N}(\mathbf{u}_t|\mathbf{a}_t,\mathbf{C}_{\boldsymbol{\omega}}(k))$ into the first term of Eq. (13).

$$\mathbb{E}_{p_{\mathrm{d}}(\mathbf{s}_{1:T},\mathbf{u}_{1:T}|k)p(k)}\left[\sum_{t=1}^{T}\mathbb{E}_{q_{\boldsymbol{\psi}}(\mathbf{a}_t|\mathbf{s}_t,\mathbf{u}_t,k)}\left[l_{\boldsymbol{\phi},\boldsymbol{\omega}}(\mathbf{s}_t,\mathbf{a}_t,\mathbf{u}_t,k) - \log q_{\boldsymbol{\psi}}(\mathbf{a}_t|\mathbf{s}_t,\mathbf{u}_t,k)\right]\right]$$

$$= \mathbb{E}_{p_{\mathrm{d}}(\mathbf{s}_{1:T},\mathbf{u}_{1:T}|k)p(k)}\left[\sum_{t=1}^{T}\mathbb{E}_{q_{\boldsymbol{\psi}}(\mathbf{a}_t|\mathbf{s}_t,\mathbf{u}_t,k)}\left[r_{\boldsymbol{\phi}}(\mathbf{s}_t,\mathbf{a}_t) - \frac{1}{2}\|\mathbf{u}_t - \mathbf{a}_t\|_{\mathbf{C}_{\boldsymbol{\omega}}^{-1}(k)}^2 - \frac{1}{2}\log|\mathbf{C}_{\boldsymbol{\omega}}(k)|\right.\right.$$

$$\left.\left. - \log q_{\boldsymbol{\psi}}(\mathbf{a}_t|\mathbf{s}_t,\mathbf{u}_t,k)\right]\right] - T d_{\mathbf{a}}\log 2\pi/2. \tag{15}$$

Lastly, by ignoring constants, Eq. (13) is equivalent to $\max_{\boldsymbol{\phi},\boldsymbol{\omega},\boldsymbol{\psi}}\min_{\boldsymbol{\theta}} \mathcal{H}(\boldsymbol{\phi},\boldsymbol{\omega},\boldsymbol{\psi},\boldsymbol{\theta})$, where

$$\mathcal{H}(\boldsymbol{\phi},\boldsymbol{\omega},\boldsymbol{\psi},\boldsymbol{\theta}) = \mathbb{E}_{p_{\mathrm{d}}(\mathbf{s}_{1:T},\mathbf{u}_{1:T}|k)p(k)}\left[\sum_{t=1}^{T}\mathbb{E}_{q_{\boldsymbol{\psi}}(\mathbf{a}_t|\mathbf{s}_t,\mathbf{u}_t,k)}\left[r_{\boldsymbol{\phi}}(\mathbf{s}_t,\mathbf{a}_t) - \frac{1}{2}\|\mathbf{u}_t - \mathbf{a}_t\|_{\mathbf{C}_{\boldsymbol{\omega}}^{-1}(k)}^2 - \log q_{\boldsymbol{\psi}}(\mathbf{a}_t|\mathbf{s}_t,\mathbf{u}_t,k)\right]\right]$$

$$- \mathbb{E}_{\widetilde{q}_{\boldsymbol{\theta}}(\mathbf{s}_{1:T},\mathbf{a}_{1:T})}\left[\sum_{t=1}^{T} r_{\boldsymbol{\phi}}(\mathbf{s}_t,\mathbf{a}_t) - \log q_{\boldsymbol{\theta}}(\mathbf{a}_t|\mathbf{s}_t)\right] + \frac{T}{2}\mathbb{E}_{p(k)}\left[\mathrm{Tr}(\mathbf{C}_{\boldsymbol{\omega}}^{-1}(k)\boldsymbol{\Sigma})\right].$$

This concludes the derivation of VILD.

As mentioned, other distributions beside the Gaussian distribution can be used for $p_{\boldsymbol{\omega}}$. For instance, let us consider a multivariate-independent Laplace distribution: $p_{\boldsymbol{\omega}}(\mathbf{u}_t|\mathbf{s}_t,\mathbf{a}_t,k) = \Pi_{d=1}^{d_{\mathbf{a}}}\frac{1}{2c_k^{(d)}}\exp(-\|\frac{\mathbf{u}_t-\mathbf{a}_t}{\mathbf{c}_k}\|_1)$, where a division of vector by vector denotes element-wise division. The Laplace distribution has heavier tails when compared to the Gaussian distribution, which makes the Laplace distribution more suitable for modeling demonstrators who tend to execute outlier actions. By using the Laplace distribution for $p_{\boldsymbol{\omega}}(\mathbf{u}_t|\mathbf{s}_t,\mathbf{a}_t,k)$, we obtain an objective

$$\mathcal{H}_{\mathrm{Lap.}} = \mathbb{E}_{p_{\mathrm{d}}(\mathbf{s}_{1:T},\mathbf{u}_{1:T},k)}\left[\sum_{t=1}^{T}\mathbb{E}_{q_{\boldsymbol{\psi}}(\mathbf{a}_t|\mathbf{s}_t,\mathbf{u}_t,k)}\left[r_{\boldsymbol{\phi}}(\mathbf{s}_t,\mathbf{a}_t) - \left\|\frac{\mathbf{u}_t - \mathbf{a}_t}{\mathbf{c}_k}\right\|_1 - \log q_{\boldsymbol{\psi}}(\mathbf{a}_t|\mathbf{s}_t,\mathbf{u}_t,k)\right]\right]$$

$$- \mathbb{E}_{\widetilde{q}_{\boldsymbol{\theta}}(\mathbf{s}_{1:T},\mathbf{a}_{1:T})}\left[\sum_{t=1}^{T} r_{\boldsymbol{\phi}}(\mathbf{s}_t,\mathbf{a}_t) - \log q_{\boldsymbol{\theta}}(\mathbf{a}_t|\mathbf{s}_t)\right] + \frac{T\sqrt{2}}{\sqrt{\pi}}\mathbb{E}_{p(k)}\left[\mathrm{Tr}(\mathbf{C}_{\boldsymbol{\omega}}^{-1}(k)\boldsymbol{\Sigma}^{1/2})\right].$$

We can see that differences between $\mathcal{H}_{\mathrm{Lap}}$ and $\mathcal{H}$ are the absolute error and scaling of the trace term.

## B   Implementation Details

We implement VILD using the PyTorch deep learning framework. For all function approximators, we use neural networks with 2 hidden-layers of 100 $\tanh$ units, except for the Humanoid task and the robosuite reaching task

---

**Algorithm 1** VILD: Variational Imitation Learning with Diverse-quality demonstrations

---

1: **Input:** Diverse-quality demonstrations $\mathcal{D}_{\mathrm{d}} = \{(\mathbf{s}_{1:T}, \mathbf{u}_{1:T}, k)_n\}_{n=1}^N$ and a replay buffer $\mathcal{B} = \varnothing$.
2: **while** Not converge **do**
3:      **while** $|\mathcal{B}| < B$ with batch size $B$ **do**          $\triangleright$ Collect samples from $\widetilde{q}_{\boldsymbol{\theta}}(\mathbf{s}_{1:T}, \mathbf{a}_{1:T})$
4:          Sample $\mathbf{a}_t \sim q_{\boldsymbol{\theta}}(\mathbf{a}_t|\mathbf{s}_t)$ and $\boldsymbol{\epsilon}_t \sim \mathcal{N}(\boldsymbol{\epsilon}_t|\mathbf{0}, \boldsymbol{\Sigma})$.
5:          Execute $\mathbf{a}_t + \boldsymbol{\epsilon}_t$, observe $\mathbf{s}'_t \sim p(\mathbf{s}'_t|\mathbf{s}_t, \mathbf{a}_t + \boldsymbol{\epsilon}_t)$, and include $(\mathbf{s}_t, \mathbf{a}_t, \mathbf{s}'_t)$ into $\mathcal{B}$
6:      Update $q_{\boldsymbol{\psi}}$ by an estimate of $\nabla_{\boldsymbol{\psi}}\mathcal{H}(\boldsymbol{\phi}, \boldsymbol{\omega}, \boldsymbol{\psi}, \boldsymbol{\theta})$.
7:      Update $p_{\boldsymbol{\omega}}$ by an estimate of $\nabla_{\boldsymbol{\omega}}\mathcal{H}(\boldsymbol{\phi}, \boldsymbol{\omega}, \boldsymbol{\psi}, \boldsymbol{\theta}) + \nabla_{\boldsymbol{\omega}}L(\boldsymbol{\omega})$.
8:      Update $r_{\boldsymbol{\phi}}$ by an estimate of $\nabla_{\boldsymbol{\phi}}\mathcal{H}_{\mathrm{IS}}(\boldsymbol{\phi}, \boldsymbol{\omega}, \boldsymbol{\psi}, \boldsymbol{\theta})$.
9:      Update $q_{\boldsymbol{\theta}}$ by an RL method (e.g., TRPO or SAC) with reward function $r_{\boldsymbol{\phi}}$.

---

where we use neural networks with 2 hidden-layers of 100 relu units. We optimize parameters $\boldsymbol{\phi}$, $\boldsymbol{\omega}$, and $\boldsymbol{\psi}$ by Adam with step-size $3 \times 10^{-4}$, $\beta_1 = 0.9$, $\beta_2 = 0.999$ and mini-batch size 256. To optimize the policy parameter $\boldsymbol{\theta}$, we use trust region policy optimization (TRPO) (Schulman et al., 2015) with batch size 1000, except on the Humanoid task where we use soft actor-critic (SAC) (Haarnoja et al., 2018) with mini-batch size 256. Note that TRPO is an on-policy RL method that uses only trajectories collected by the current policy, while SAC is an off-policy RL method that use trajectories collected by previous policies. On-policy methods are generally more stable than off-policy methods, while off-policy methods are generally more data-efficient (Gu et al., 2017). We use SAC for Humanoid mainly due to its high data-efficiency. When SAC is used, we also use trajectories collected by previous policies to approximate the expectation over the trajectory density $\tilde{q}_{\boldsymbol{\theta}}(\mathbf{s}_{1:T}, \mathbf{a}_{1:T})$.

For the distribution $p_{\boldsymbol{\omega}}(\mathbf{u}_t|\mathbf{s}_t, \mathbf{a}_t, k) = \mathcal{N}(\mathbf{u}_t|\mathbf{a}_t, \mathbf{C}_{\boldsymbol{\omega}}(k))$, we use diagonal covariances $\mathbf{C}_{\boldsymbol{\omega}}(k) = \mathrm{diag}(\mathbf{c}_k)$, where $\boldsymbol{\omega} = \{\mathbf{c}_k\}_{k=1}^K$ and $\mathbf{c}_k \in \mathbb{R}_+^{d_\mathbf{a}}$ are parameter vectors to be learned. For the distribution $q_{\boldsymbol{\psi}}(\mathbf{a}_t|\mathbf{s}_t, \mathbf{u}_t, k)$, we use a Gaussian distribution with diagonal covariance, where the mean and logarithm of the standard deviation are the outputs of neural networks. Since $k$ is a discrete variable, we represent $q_{\boldsymbol{\psi}}(\mathbf{a}_t|\mathbf{s}_t, \mathbf{u}_t, k)$ by neural networks that have $K$ output heads and take input vectors $(\mathbf{s}_t, \mathbf{u}_t)$; The $k$-th output head corresponds to (the mean and log-standard-deviation of) $q_{\boldsymbol{\psi}}(\mathbf{a}_t|\mathbf{s}_t, \mathbf{u}_t, k)$. We also pre-train the mean function of $q_{\boldsymbol{\psi}}(\mathbf{a}_t|\mathbf{s}_t, \mathbf{u}_t, k)$, by performing least-squares regression for 1000 gradient steps with target value $\mathbf{u}_t$. This pre-training is done to obtain reasonable initial predictions. For the policy $q_{\boldsymbol{\theta}}(\mathbf{a}_t|\mathbf{s}_t)$, we use a Gaussian policy with diagonal covariance, where the mean and logarithm of the standard deviation are outputs of neural networks. We use $\boldsymbol{\Sigma} = 10^{-8}\boldsymbol{I}$ in experiments.

To control exploration-exploitation trade-off, we use an entropy coefficient $\alpha = 0.0001$ in TRPO. In SAC, the value of $\alpha$ is optimized so that the policy has a certain value of entropy, as described by Haarnoja et al. (2018). Note that including $\alpha$ in VILD is equivalent to rescaling quantities in the model by $\alpha$, i.e., $\exp(r_{\boldsymbol{\phi}}(\mathbf{s}_t, \mathbf{a}_t)/\alpha)$ and $(p_{\boldsymbol{\omega}}(\mathbf{u}_t|\mathbf{s}_t, \mathbf{a}_t, k))^{\frac{1}{\alpha}}$. A discount factor $0 < \gamma < 1$ may be included similarly, and we use $\gamma = 0.99$ in experiments.

For all methods, we regularize the reward/discriminator function by the gradient penalty (Gulrajani et al., 2017) with coefficient 10, since it was previously shown to improve performance of generative adversarial learning methods. For methods that learn a reward function, namely VILD, AIRL, and MaxEnt-IRL, we apply a sigmoid function to the output of a reward network to bound reward values. We found that without the bounds, reward values of the agent can be highly negative in the early stage of learning, which makes RL methods prematurely converge to poor policies. An explanation of this phenomenon is that, in MDPs with large state and action spaces, distribution of demonstrations and distribution of agent's trajectories are not overlapped in the early stage of learning. In such a scenario, it is trivial to learn a reward function which tends to positive-infinity values for demonstrations and negative-infinity values for agent's trajectories. While the gradient penalty regularizer slightly remedies this issue, we found that the regularizer alone is insufficient to prevent this scenario. Moreover, for VILD, it is beneficial to bound the reward function to control a trade-off between the immediate reward and the squared error when optimizing $\boldsymbol{\psi}$.

A pseudo-code of VILD with IS is given in Algorithm 1, where the reward parameter is updated by IS gradient in line 8. For VILD without IS, the reward parameter is instead updated by an estimate of $\nabla_{\boldsymbol{\phi}}\mathcal{H}(\boldsymbol{\phi}, \boldsymbol{\omega}, \boldsymbol{\psi}, \boldsymbol{\theta})$. The regularizer $L(\boldsymbol{\omega}) = T\mathbb{E}_{p(k)}[\log|\mathbf{C}_{\boldsymbol{\omega}}^{-1}(k)|]/2$ penalizes large value of $\mathbf{C}_{\boldsymbol{\omega}}(k)$. A source-code of our implementation will be publicly available.

## C  EXPERIMENT DETAILS

In this section, we describe experimental settings and data generation. We also give brief reviews of methods compared against VILD in the experiments.

Table 2: Performance of a random policy $\pi_0$, the optimal policy $\pi^\star$, and demonstrators with the Gaussian noisy policy.

| $\sigma_k$ | Cheetah | Ant | Walker | Humanoid |
|---|---|---|---|---|
| $(\pi_0)$ | -0.58 | 995 | 131 | 222 |
| $(\pi^\star)$ | 4624 | 4349 | 4963 | 5093 |
| 0.01 | 4311 | 3985 | 4434 | 4315 |
| 0.05 | 3978 | 3861 | 3486 | 5140 |
| 0.01 | 4019 | 3514 | 4651 | 5189 |
| 0.25 | 1853 | 536 | 4362 | 3628 |
| 0.40 | 1090 | 227 | 467 | 5220 |
| 0.6 | 567 | -73 | 523 | 2593 |
| 0.7 | 267 | -208 | 332 | 1744 |
| 0.8 | -45 | -979 | 283 | 735 |
| 0.9 | -399 | -328 | 255 | 538 |
| 1.0 | -177 | -203 | 249 | 361 |

Table 3: Performance of a random policy $\pi_0$, the optimal policy $\pi^\star$, and demonstrators with the TSD noisy policy.

| $\sigma_k$ | Cheetah | Ant | Walker | Humanoid |
|---|---|---|---|---|
| $(\pi_0)$ | -0.58 | 995 | 131 | 222 |
| $(\pi^\star)$ | 4624 | 4349 | 4963 | 5093 |
| 0.01 | 4362 | 3758 | 4695 | 5130 |
| 0.05 | 4015 | 3623 | 4528 | 5099 |
| 0.01 | 3741 | 3368 | 2362 | 5195 |
| 0.25 | 1301 | 873 | 644 | 1675 |
| 0.40 | -203 | 231 | 302 | 610 |
| 0.6 | -230 | -51 | 29 | 249 |
| 0.7 | -249 | -37 | 24 | 221 |
| 0.8 | -416 | -567 | 14 | 191 |
| 0.9 | -389 | -751 | 7 | 178 |
| 1.0 | -424 | -269 | 4 | 169 |

## C.1 EXPERIMENTAL SETTING AND DATA GENERATION FOR BENCHMARK TASKS

For the benchmark experiment in Section 5.1, we evaluate VILD on four continuous-control benchmark tasks from OpenAI gym platform (Brockman et al., 2016) with the Mujoco physics simulator: HalfCheetah, Ant, Walker2d, and Humanoid. To obtain the optimal policy for generating demonstrations, we use the ground-truth reward function of each task to pre-train $\pi^\star$ with TRPO. We generate diverse-quality demonstrations by using $K = 10$ demonstrators according to the graphical model in Figure 1(b). We consider two types of the noisy policy $p(\mathbf{u}_t|\mathbf{s}_t, \mathbf{a}_t, k)$: a Gaussian noisy policy and a time-signal-dependent (TSD) noisy policy.

**Gaussian noisy policy.** We use a Gaussian noisy policy $\mathcal{N}(\mathbf{u}_t|\mathbf{a}_t, \sigma_k^2 \mathbf{I})$ with a constant covariance. The value of $\sigma_k$ for each of the 10 demonstrators is $0.01, 0.05, 0.1, 0.25, 0.4, 0.6, 0.7, 0.8, 0.9$ and $1.0$, respectively. Note that our model assumption on $p_{\boldsymbol{\omega}}$ corresponds to this Gaussian noisy policy. Table 2 shows the performance of demonstrators (in terms of cumulative ground-truth rewards) with this Gaussian noisy policy. A random policy $\pi_0$ is an initial policy neural network for learning; The network weights are initialized such that the magnitude of actions is small. Note that this initialization scheme is a common practice in deep RL (Gu et al., 2017).

**TSD noisy policy.** To make learning more challenging, we generate demonstrations according to a noise characteristic of human motor control, where a magnitude of noises is proportion to a magnitude of actions and increases with execution time (van Beers et al., 2004). Specifically, we generate demonstrations using a Gaussian distribution $\mathcal{N}(\mathbf{u}_t|\mathbf{a}_t, \text{diag}(\mathbf{b}_k(t) \times \|\mathbf{a}_t\|_1/d_{\mathbf{a}}))$, where the covariance is proportion to the magnitude of action and depends on time steps and $\times$ denotes an element-wise product. We call this policy time-signal-dependent (TSD) noisy policy. Here, $\mathbf{b}_k(t)$ is a sample of a noise process whose noise variance increases over time, as shown in Figure 6. We obtain this noise process for the $k$-th demonstrator by reversing Ornstein–Uhlenbeck (OU) processes with parameters $\theta = 0.15$ and $\sigma = \sigma_k$ (Uhlenbeck & Ornstein, 1930)[4]. The value of $\sigma_k$ for each demonstrator is $0.01, 0.05, 0.1, 0.25, 0.4, 0.6, 0.7, 0.8, 0.9$, and $1.0$, respectively. Table 3 shows the performance of demonstrators with this TSD noisy policy. Learning from demonstrations generated by TSD is challenging; The Gaussian model of $p_{\boldsymbol{\omega}}$ cannot perfectly model the TSD noisy policy, since the ground-truth variance is a function of actions and time steps.

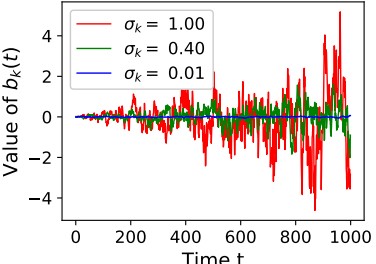

Figure 6: Samples $\mathbf{b}_k(t)$ drawn from noise processes used for the TSD noisy policy.

## C.2 EXPERIMENTAL SETTING FOR ROBOSUITE REACHING TASK

For the real-world data experiment in Section 5.2, we use a robot control task from the robosuite environment Fan et al. (2018) and a crowdsourced demonstration dataset from Mandlekar et al. (2018)[5]. These demonstrations are collected for object-manipulation tasks such as assembly tasks. These object-manipulation tasks require

---

[4]OU process is commonly used to generate time-correlated noises where the noise variance decays towards zero. We reserve this process along the time axis, so that the noise variance grows over time.

[5]We use the publicly available dataset: http://roboturk.stanford.edu/dataset.html

the agent to perform three subtasks: reaching, picking, and placing. In our preliminary experiments, none of IL methods successfully learns object-manipulation policies, since the agent often fails at picking the object. We expect that a hierarchical policy is necessary to perform these manipulation tasks, due to the hierarchical structure (i.e., subtasks) of these tasks. Since hierarchical IL is not in the scope of this paper, we consider the subtask of reaching where non-hierarchical policies suffice. We leave an extension of VILD to hierarchical policy for future work.

In this experiment, we consider the subtask of reaching, which is still challenging for IL due to diverse quality of crowdsourced demonstrations. To obtain reaching demonstrations from the original object-manipulation demonstrations (we use the *SawyerNutAssemblyRound* dataset), we terminate demonstrations after the robot's end-effector contacts the target object. After applying such a termination procedure, the dataset used in this experiment consists of 10 randomly chosen demonstrations ($N = 10$) whose length $T$ is approximately 500 time steps. The number of state-action pairs in this demonstration dataset is approximately 5000. Since we do not know the actual number of demonstrators that collected these $N = 10$ demonstrations, we use the strategy described in Section 3.3; we set $K = N$ and $k = n$. We use true states of the robot and do not use visual observations. Since the reaching task does not require picking the object, we disable the gripper control command of the robot. The state space of this task is $\mathcal{S} \subseteq \mathbb{R}^{44}$, and the action space of this task is $\mathcal{A} \subseteq \mathbb{R}^{7}$. Figure 11 shows three examples of demonstrations used in this experiment. We can notice the differences in qualities of demonstrations, e.g., demonstration 3 is better than demonstration 2 since the robot reaches the object faster.

The performance of learned policies are evaluated using a reward function whose values are inverse proportion to the distance between the object and the end-effector (i.e., small distance yields high reward). We repeat the experiment for 5 trials using the same dataset and report the average performance (undiscounted cumulative rewards). For each trial, we generate 100 test trajectories for evaluating the performance. Note that the number of test trajectories in this experiment is larger than that in the benchmark experiments. This is because the initial states of this reaching task is much more varied than those in benchmark tasks. We do not evaluate VILD without IS and VAIL, since in benchmarks VILD with IS performs better than VILD without IS and VAIL is comparable to GAIL.

For all methods, we use neural networks with 2 hidden-layers of 100 relu units. We update policy parameters by TRPO with the same hyper-parameters as the benchmark experiments. We pre-train the mean of Gaussian policies for all methods by behavior cloning (i.e., we apply 1000 gradient descent steps of least-squares regression). To pre-train InfoGAIL which learns a context-dependent policy, we use the variable $k$ as context for pre-training. For VILD, we apply the log-sigmoid function to the reward function. Specifically, we parameterize the reward function as $r_{\phi}(\mathbf{s}, \mathbf{a}) = \log D_{\phi}(\mathbf{s}, \mathbf{a})$ where $D_{\phi}(\mathbf{s}, \mathbf{a}) = \frac{\exp(d_{\phi}(\mathbf{s}, \mathbf{a}))}{\exp(d_{\phi}(\mathbf{s}, \mathbf{a}))+1}$ and $d_{\phi} : \mathcal{S} \times \mathcal{A} \to \mathbb{R}$. We also apply a substitution $-\log D_{\phi}(\mathbf{s}, \mathbf{a}) \to \log(1 - D_{\phi}(\mathbf{s}, \mathbf{a}))$, which is a common practice in GAN literature (Fedus et al., 2018). By doing so, we obtain an objective of VILD that closely resembles the objective of GAIL:

$$\mathcal{H}_{\log}(\boldsymbol{\phi}, \boldsymbol{\omega}, \boldsymbol{\psi}, \boldsymbol{\theta}) = \mathbb{E}_{p_{\mathrm{d}}(\mathbf{s}_{1:T}, \mathbf{u}_{1:T}|k)p(k)} \Big[ \sum_{t=1}^{T} \mathbb{E}_{q_{\boldsymbol{\psi}}(\mathbf{a}_t|\mathbf{s}_t, \mathbf{u}_t, k)} \Big[ \log D_{\phi}(\mathbf{s}, \mathbf{a}) - \tfrac{1}{2} \| \mathbf{u}_t - \mathbf{a}_t \|^2_{\mathbf{C}_{\boldsymbol{\omega}}^{-1}(k)} \Big] + H_t(q_{\boldsymbol{\psi}}) \Big]$$
$$+ \mathbb{E}_{\tilde{q}_{\boldsymbol{\theta}}(\mathbf{s}_{1:T}, \mathbf{a}_{1:T})} \Big[ \sum_{t=1}^{T} \log(1 - D_{\phi}(\mathbf{s}, \mathbf{a})) + \log q_{\boldsymbol{\theta}}(\mathbf{a}_t|\mathbf{s}_t) \Big] + \tfrac{T}{2} \mathbb{E}_{p(k)} \Big[ \mathrm{Tr}(\mathbf{C}_{\boldsymbol{\omega}}^{-1}(k)\boldsymbol{\Sigma}) \Big]. \tag{16}$$

We use this variant of VILD in this experiment since it performs better than VILD with the standard reward function. Although we omit the IS distribution in this equation for clarity, we use IS in this experiment.

## C.3 COMPARISON METHODS

Here, we briefly review methods compared against VILD in our experiments. We firstly review online IL methods, which learn a policy by RL and require additional transition samples from MDPs.

**MaxEnt-IRL.** Maximum (causal) entropy IRL (MaxEnt-IRL) (Ziebart et al., 2010) is a well-known IRL method. The original derivation of the method is based on the maximum entropy principle (Jaynes, 1957) but for causal interactions, and uses a linear-in-parameter reward function: $r_{\phi}(\mathbf{s}_t, \mathbf{a}_t) = \boldsymbol{\phi}^{\top} b(\mathbf{s}_t, \mathbf{a}_t)$ with a basis function $b$. Here, we consider an alternative derivation which is applicable to non-linear reward function (Finn et al., 2016). Briefly speaking, MaxEnt-IRL learns a reward parameter by minimizing a KL divergence from a data distribution $p^{\star}(\mathbf{s}_{1:T}, \mathbf{a}_{1:T})$ to a model $p_{\phi}(\mathbf{s}_{1:T}, \mathbf{a}_{1:T}) = \frac{1}{Z_{\phi}} p_1(\mathbf{s}_1) \Pi_{t=1}^{T} p(\mathbf{s}_{t+1}|\mathbf{s}_t, \mathbf{a}_t) \exp(r_{\phi}(\mathbf{s}_t, \mathbf{a}_t)/\alpha)$, where $Z_{\phi}$ is the normalization term. Minimizing this KL divergence is equivalent to solving $\max_{\phi} \mathbb{E}_{p^{\star}(\mathbf{s}_{1:T}, \mathbf{a}_{1:T})} \big[ \Sigma_{t=1}^{T} r_{\phi}(\mathbf{s}_t, \mathbf{a}_t) \big] - \log Z_{\phi}$. To compute $\log Z_{\phi}$, we can use the importance sampling approach (Finn et al., 2016) or the variational approache as done in VILD. The latter leads to a max-min problem

$$\max_{\boldsymbol{\phi}} \min_{\boldsymbol{\theta}} \mathbb{E}_{p^{\star}(\mathbf{s}_{1:T}, \mathbf{a}_{1:T})} \Big[ \Sigma_{t=1}^{T} r_{\phi}(\mathbf{s}_t, \mathbf{a}_t) \Big] - \mathbb{E}_{q_{\boldsymbol{\theta}}(\mathbf{s}_{1:T}, \mathbf{a}_{1:T})} \Big[ \Sigma_{t=1}^{T} r_{\phi}(\mathbf{s}_t, \mathbf{a}_t) - \alpha \log q_{\boldsymbol{\theta}}(\mathbf{a}_t|\mathbf{s}_t) \Big],$$

where $q_{\boldsymbol{\theta}}(\mathbf{s}_{1:T}, \mathbf{a}_{1:T}) = p_1(\mathbf{s}_1)\Pi_{t=1}^{T} p(\mathbf{s}_{t+1}|\mathbf{s}_t, \mathbf{a}_t) q_{\boldsymbol{\theta}}(\mathbf{a}_t|\mathbf{s}_t)$. The policy $q_{\boldsymbol{\theta}}(\mathbf{a}_t|\mathbf{s}_t)$ maximizes the learned reward function and is the solution of IL.

As we mentioned, the proposed model in VILD is based on the model of MaxEnt-IRL. By comparing the max-min problem of MaxEnt-IRL and the max-min problem of VILD, we can see that the main difference are the variational distribution $q_{\boldsymbol{\psi}}$ and the noisy policy model $p_{\boldsymbol{\omega}}$. If we assume that $q_{\boldsymbol{\psi}}$ and $p_{\boldsymbol{\omega}}$ are Dirac delta functions: $q_{\boldsymbol{\psi}}(\mathbf{a}_t|\mathbf{s}_t, \mathbf{u}_t, k) = \delta_{\mathbf{a}_t=\mathbf{u}_t}$ and $p_{\boldsymbol{\omega}}(\mathbf{u}_t|\mathbf{a}_t, \mathbf{s}_t, k) = \delta_{\mathbf{u}_t=\mathbf{a}_t}$, then the max-min problem of VILD reduces to the max-min problem of MaxEnt-IRL. In other words, if we assume that all demonstrators execute the optimal policy and have an equal level of expertise, then VILD reduces to MaxEnt-IRL.

**GAIL.** Generative adversarial IL (GAIL) (Ho & Ermon, 2016) performs occupancy measure matching via generative adversarial networks (GAN) to learn the optimal policy from expert demonstrations. Specifically, GAIL finds a parameterized policy $\pi_{\boldsymbol{\theta}}$ such that the occupancy measure $\rho_{\pi_{\boldsymbol{\theta}}}(\mathbf{s}, \mathbf{a})$ of $\pi_{\boldsymbol{\theta}}$ is similar to the occupancy measure $\rho_{\pi^\star}(\mathbf{s}, \mathbf{a})$ of $\pi^\star$. Here, $\rho_{\pi}(\mathbf{s}, \mathbf{a}) = \mathbb{E}_{p_{\pi}(\mathbf{s}_{1:T}, \mathbf{a}_{1:T})}[\Sigma_{t=0}^{T}\delta(\mathbf{s}_t - \mathbf{s}, \mathbf{a}_t - \mathbf{a})]$ is the state-action occupancy measure of $\pi$ and satisfies the equality $\mathbb{E}_{p_{\pi}(\mathbf{s}_{1:T}, \mathbf{a}_{1:T})}[\Sigma_{t=1}^{T} r(\mathbf{s}_t, \mathbf{a}_t)] = \iint_{\mathcal{S} \times \mathcal{A}} \rho_{\pi}(\mathbf{s}, \mathbf{a}) r(\mathbf{s}, \mathbf{a}) \mathrm{d}\mathbf{s}\mathrm{d}\mathbf{a} = \mathbb{E}_{\pi}[r(\mathbf{s}, \mathbf{a})]$. To measure the similarity, GAIL uses the Jensen-Shannon divergence, which is estimated and minimized by the following generative-adversarial training objective:

$$\min_{\boldsymbol{\theta}} \max_{\boldsymbol{\phi}} \mathbb{E}_{\rho_{\pi^\star}}\left[\log D_{\boldsymbol{\phi}}(\mathbf{s}, \mathbf{a})\right] + \mathbb{E}_{\rho_{\pi_{\boldsymbol{\theta}}}}\left[\log(1 - D_{\boldsymbol{\phi}}(\mathbf{s}, \mathbf{a})) + \alpha \log \pi_{\boldsymbol{\theta}}(\mathbf{a}_t|\mathbf{s}_t)\right],$$

where $D_{\boldsymbol{\phi}}(\mathbf{s}, \mathbf{a}) = \frac{\exp(d_{\boldsymbol{\phi}}(\mathbf{s}, \mathbf{a}))}{\exp(d_{\boldsymbol{\phi}}(\mathbf{s}, \mathbf{a})) + 1}$ is called a discriminator. The minimization problem w.r.t. $\boldsymbol{\theta}$ is achieved using RL with a reward function $-\log(1 - D_{\boldsymbol{\phi}}(\mathbf{s}, \mathbf{a}))$.

**AIRL.** Adversarial IRL (AIRL) (Fu et al., 2018) was proposed to overcome a limitation of GAIL regarding reward function: GAIL does not learn the expert reward function, since GAIL has $D_{\boldsymbol{\phi}}(\mathbf{s}, \mathbf{a}) = 0.5$ at the saddle point for every states and actions. To overcome this limitation while taking advantage of generative-adversarial training, AIRL learns a reward function by solving

$$\max_{\boldsymbol{\phi}} \mathbb{E}_{p^\star(\mathbf{s}_{1:T}, \mathbf{a}_{1:T})}\left[\Sigma_{t=1}^{T} \log D_{\boldsymbol{\phi}}(\mathbf{s}, \mathbf{a})\right] + \mathbb{E}_{q_{\boldsymbol{\theta}}(\mathbf{s}_{1:T}, \mathbf{a}_{1:T})}\left[\Sigma_{t=1}^{T} \log(1 - D_{\boldsymbol{\phi}}(\mathbf{s}, \mathbf{a}))\right],$$

where $D_{\boldsymbol{\phi}}(\mathbf{s}, \mathbf{a}) = \frac{\exp(r_{\boldsymbol{\phi}}(\mathbf{s}, \mathbf{a}))}{\exp(r_{\boldsymbol{\phi}}(\mathbf{s}, \mathbf{a})) + q_{\boldsymbol{\theta}}(\mathbf{a}|\mathbf{s})}$. The policy $q_{\boldsymbol{\theta}}(\mathbf{a}_t|\mathbf{s}_t)$ is learned by RL with a reward function $r_{\boldsymbol{\phi}}(\mathbf{s}_t, \mathbf{a}_t)$. Fu et al. (2018) showed that the gradient of this objective w.r.t. $\boldsymbol{\phi}$ is equivalent to the gradient of MaxEnt-IRL w.r.t. $\boldsymbol{\phi}$. The authors also proposed an approach to disentangle reward function, which leads to a better performance in transfer learning settings. Nonetheless, this disentangle approach is general and can be applied to other IRL methods, including MaxEnt-IRL and VILD. We do not evaluate AIRL with disentangle reward function.

We note that, based on the relation between MaxEnt-IRL and VILD, we can extend VILD to use a training procedure of AIRL. Specifically, by applying the same derivation from MaxEnt-IRL to AIRL by Fu et al. (2018), we can derive a variant of VILD which learns a reward parameter by solving $\max_{\boldsymbol{\phi}} \mathbb{E}_{p_{\mathrm{d}}(\mathbf{s}_{1:T}, \mathbf{u}_{1:T}|k)p(k)}[\Sigma_{t=1}^{T} \mathbb{E}_{q_{\boldsymbol{\psi}}(\mathbf{a}_t|\mathbf{s}_t, \mathbf{u}_t, k)}[\log D_{\boldsymbol{\phi}}(\mathbf{s}, \mathbf{a})]] + \mathbb{E}_{\tilde{q}_{\boldsymbol{\theta}}(\mathbf{s}_{1:T}, \mathbf{a}_{1:T})}[\Sigma_{t=1}^{T} \log(1 - D_{\boldsymbol{\phi}}(\mathbf{s}, \mathbf{a}))]$. We do not evaluate this variant of VILD in our experiment.

**VAIL.** Variational adversarial IL (VAIL) (Peng et al., 2019) improves upon GAIL by using variational information bottleneck (VIB) (Alemi et al., 2017). VIB aims to compress information flow by minimizing a variational bound of mutual information. This compression filters irrelevant signals, which leads to less over-fitting. To achieve this in GAIL, VAIL learns the discriminator $D_{\boldsymbol{\phi}}$ by an optimization problem

$$\min_{\boldsymbol{\phi}, E} \max_{\beta \geqslant 0} \mathbb{E}_{\rho_{\pi^\star}}\left[\mathbb{E}_{E(\mathbf{z}|\mathbf{s}, \mathbf{a})}\left[-\log D_{\boldsymbol{\phi}}(\mathbf{z})\right]\right] + \mathbb{E}_{\rho_{\pi_{\boldsymbol{\theta}}}}\left[\mathbb{E}_{E(\mathbf{z}|\mathbf{s}, \mathbf{a})}\left[-\log(1 - D_{\boldsymbol{\phi}}(\mathbf{z}))\right]\right]$$
$$+ \beta \mathbb{E}_{(\rho_{\pi^\star} + \rho_{\pi_{\boldsymbol{\theta}}})/2}\left[\mathrm{KL}(E(\mathbf{z}|\mathbf{s}, \mathbf{a})|p(\mathbf{z})) - I_c\right],$$

where $\mathbf{z}$ is an encode vector, $E(\mathbf{z}|\mathbf{s}, \mathbf{a})$ is an encoder, $p(\mathbf{z})$ is a prior distribution of $\mathbf{z}$, $I_c$ is the target value of mutual information, and $\beta > 0$ is a Lagrange multiplier. With this discriminator, the policy $\pi_{\boldsymbol{\theta}}(\mathbf{a}_t|\mathbf{s}_t)$ is learned by RL with a reward function $-\log(1 - D_{\boldsymbol{\phi}}(\mathbb{E}_{E(\mathbf{z}|\mathbf{s}, \mathbf{a})}[\mathbf{z}]))$.

It might be expected that the compression may make VAIL robust against diverse-quality demonstrations, since irrelevant signals in low-quality demonstrations are filtered out via the encoder. However, we find that this is not the case, and VAIL does not improve much upon GAIL in our experiments. This is perhaps because VAIL compress information from both demonstrators and agent's trajectories. Meanwhile in our setting, irrelevant signals are generated only by demonstrators. Therefore, the information bottleneck may also filter out relevant signals in agent's trajectories, which lead to poor performances.

**InfoGAIL.** Information maximizing GAIL (InfoGAIL) (Li et al., 2017) is an extension of GAIL for learning a multi-modal policy in MM-IL. The key idea of InfoGAIL is to introduce a context variable $z$ to the GAIL formulation and learn a context-dependent policy $\pi_{\boldsymbol{\theta}}(\mathbf{a}|\mathbf{s}, z)$, where each context represents each mode of the

multi-modal policy. To ensure that the context is not ignored during learning, InfoGAIL regularizes GAIL's objective so that a mutual information between contexts and state-action variables is maximized. This mutual information is indirectly maximized via maximizing a variational lower-bound of mutual information. By doing so, InfoGAIL solves a min-max problem

$$\min_{\boldsymbol{\theta}, Q} \max_{\boldsymbol{\phi}} \mathbb{E}_{\rho_{\pi^\star}} \left[ \log D_{\boldsymbol{\phi}}(\mathbf{s}, \mathbf{a}) \right] + \mathbb{E}_{\rho_{\pi_{\boldsymbol{\theta}}}} \left[ \log(1 - D_{\boldsymbol{\phi}}(\mathbf{s}, \mathbf{a})) + \alpha \log \pi_{\boldsymbol{\theta}}(\mathbf{a}|\mathbf{s}, z) \right] + \lambda L(\pi_{\boldsymbol{\theta}}, Q),$$

where $L(\pi_{\boldsymbol{\theta}}, Q) = \mathbb{E}_{p(z)\pi_{\boldsymbol{\theta}}(\mathbf{a}|\mathbf{s}, z)} \left[ \log Q(z|\mathbf{s}, \mathbf{a}) - \log p(z) \right]$ is a lower-bound of mutual information, $Q(z|\mathbf{s}, \mathbf{a})$ is an encoder neural network, and $p(z)$ is a prior distribution of contexts. In our experiment, the number of context $z$ is set to be the number of demonstrators $K$. As discussed in Section 1, when knowing the level of demonstrators' expertise, we may choose contexts that correspond to high-expertise demonstrator. In other words, we may hand-craft the prior distribution $p(z)$ so that a probability of contexts is proportion to the level of demonstrators' expertise. Nonetheless, for fair comparison, we do not use the oracle knowledge about the level of demonstrators' expertise, and set $p(z)$ to be a uniform distribution. For the Humanoid task in our experiment, we use the Wasserstein-distance variant of InfoGAIL (Li et al., 2017), since the Jensen-Shannon-divergence variant does not perform well in this task.

Next, we review offline IL methods. These methods learn a policy based on supervised learning and do not require additional transition samples from MDPs.

**BC.** Behavior cloning (BC) (Pomerleau, 1988) is perhaps the simplest IL method. BC treats an IL problem as a standard supervised learning problem and ignores dependency between states distributions and policy. For continuous action space, BC solves a least-square regression problem to learn a parameter $\boldsymbol{\theta}$ of a deterministic policy $\pi_{\boldsymbol{\theta}}(\mathbf{s}_t)$:

$$\min_{\boldsymbol{\theta}} \mathbb{E}_{p^\star(\mathbf{s}_{1:T}, \mathbf{a}_{1:T})} \left[ \sum_{t=1}^{T} \|\mathbf{a}_t - \pi_{\boldsymbol{\theta}}(\mathbf{s}_t)\|_2^2 \right].$$

**BC-D.** BC with Diverse-quality demonstrations (BC-D) is a simple extension of BC for handling diverse-quality demonstrations. This method is based on the naive model in Section 3.1, and we consider it mainly for evaluation purpose. BC-D uses supervised learning to learn a policy parameter $\boldsymbol{\theta}$ and expertise parameter $\boldsymbol{\omega}$ of a model $p_{\boldsymbol{\theta}, \boldsymbol{\omega}}(\mathbf{s}_{1:T}, \mathbf{u}_{1:T}, k) = p(k)p(\mathbf{s}_1)\Sigma_{t=1}^{T} p(\mathbf{s}_{t+1}|\mathbf{s}_t, \mathbf{u}_t) \int_{\mathcal{A}} \pi_{\boldsymbol{\theta}}(\mathbf{a}_t|\mathbf{s}_t) p_{\boldsymbol{\omega}}(\mathbf{u}_t|\mathbf{s}_t, \mathbf{a}_t, k) \mathrm{d}\mathbf{a}_t$. To learn the parameters, we minimize the KL divergence from data distribution to the model. By using the variational approach to handle integration over the action space, BC-D solves an optimization problem

$$\max_{\boldsymbol{\theta}, \boldsymbol{\omega}, \boldsymbol{\nu}} \mathbb{E}_{p_{\mathrm{d}}(\mathbf{s}_{1:T}, \mathbf{u}_{1:T}|k)p(k)} \left[ \sum_{t=1}^{T} \mathbb{E}_{q_{\boldsymbol{\nu}}(\mathbf{a}_t|\mathbf{s}_t, \mathbf{u}_t, k)} \left[ \log \frac{\pi_{\boldsymbol{\theta}}(\mathbf{a}_t|\mathbf{s}_t) p_{\boldsymbol{\omega}}(\mathbf{u}_t|\mathbf{s}_t, \mathbf{a}_t, k)}{q_{\boldsymbol{\nu}}(\mathbf{a}_t|\mathbf{s}_t, \mathbf{u}_t, k)} \right] \right],$$

where $q_{\boldsymbol{\nu}}(\mathbf{a}_t|\mathbf{s}_t, \mathbf{u}_t, k)$ is a variational distribution with parameters $\boldsymbol{\nu}$. We note that the model $p_{\boldsymbol{\theta}, \boldsymbol{\omega}}(\mathbf{s}_{1:T}, \mathbf{u}_{1:T}, k)$ of BC-D can be regarded as a regression-extension of the two-coin model proposed by Raykar et al. (2010) for classification with noisy labels.

**Co-teaching.** Co-teaching (Han et al., 2018) is the state-of-the-art method to perform classification with noisy labels. This method trains two neural networks such that mini-batch samples are exchanged under a small loss criteria. We extend this method to learn a policy by least-square regression. Specifically, let $\pi_{\boldsymbol{\theta}_1}(\mathbf{s}_t)$ and $\pi_{\boldsymbol{\theta}_2}(\mathbf{s}_t)$ be two neural networks representing policies, and $\nabla_{\boldsymbol{\theta}} L(\boldsymbol{\theta}, \mathcal{B}) = \nabla_{\boldsymbol{\theta}} \Sigma_{(\mathbf{s}, \mathbf{a}) \in \mathcal{B}} \|\mathbf{a} - \pi_{\boldsymbol{\theta}}(\mathbf{s})\|_2^2$ be gradients of a least-square loss estimated by using a mini-batch $\mathcal{B}$. The parameters $\boldsymbol{\theta}_1$ and $\boldsymbol{\theta}_2$ are updated by iterates:

$$\boldsymbol{\theta}_1 \leftarrow \boldsymbol{\theta}_1 - \eta \nabla_{\boldsymbol{\theta}_1} L(\boldsymbol{\theta}_1, \mathcal{B}_{\boldsymbol{\theta}_2}), \qquad \boldsymbol{\theta}_2 \leftarrow \boldsymbol{\theta}_2 - \eta \nabla_{\boldsymbol{\theta}_2} L(\boldsymbol{\theta}_2, \mathcal{B}_{\boldsymbol{\theta}_1}).$$

The mini-batch $\mathcal{B}_{\boldsymbol{\theta}_2}$ for updating $\boldsymbol{\theta}_1$ is obtained such that $\mathcal{B}_{\boldsymbol{\theta}_2}$ incurs small loss when using prediction from $\pi_{\boldsymbol{\theta}_2}$, i.e., $\mathcal{B}_{\boldsymbol{\theta}_2} = \mathrm{argmin}_{\mathcal{B}'} L(\boldsymbol{\theta}_2, \mathcal{B}')$. Similarly, the mini-batch $\mathcal{B}_{\boldsymbol{\theta}_1}$ for updating $\boldsymbol{\theta}_2$ is obtained such that $\mathcal{B}_{\boldsymbol{\theta}_1}$ incurs small loss when using prediction from $\pi_{\boldsymbol{\theta}_1}$. For evaluating the performance, we use the policy network $\pi_{\boldsymbol{\theta}_1}$.

## D  ADDITIONAL EXPERIMENTAL RESULTS

**Results against online IL methods.** Figure 7 shows the learning curves of VILD and existing online IL methods against the number of transition samples. It can be seen that for both types of noisy policy, VILD with and without IS outperform existing methods overall, except on the Humanoid tasks where most methods achieve comparable performance.

**Results against offline IL methods.** Figure 8 shows learning curves of offline IL methods, namely BC, BC-D, and Co-teaching. For comparison, the figure also shows the final performance of VILD with and without IS, according to Table 1. We can see that these offline methods do not perform well, especially on the high-dimensional Humanoid task. The poor performance of these methods is due to the issues of compounding error and low-quality demonstrations. Specifically, BC performs the worst, since it suffers from both issues. Still, BC may learn well in the early stage of learning, but its performance sharply degrades, as seen in Ant and Walker2d. This phenomena can be explained as an empirical effect of memorization in deep neural networks (Arpit et al.,

Table 4: Performance in the last iterations in terms of the mean and standard error of cumulative rewards over 5 trials (higher is better) in the robosuite reaching task. Boldfaces indicate best and comparable methods according to t-test with significance level 0.01.

| VILD (IS) | AIRL | GAIL | MaxEnt-IRL | InfoGAIL | InfoGAIL (best context) |
|---|---|---|---|---|---|
| **20.84 (1.17)** | 6.44 (0.40) | 11.40 (0.64) | **17.18 (1.09)** | 10.61 (0.52) | 11.52 (1.93) |

2017). Namely, deep neural networks learn to remember samples with simple patterns first (i.e., high-quality demonstrations from experts), but as learning progresses the networks overfit to samples with difficult patterns (i.e., low-quality demonstrations from amateurs). Co-teaching is the-state-of-the-art method to avoid this effect, and we can see that it performs significantly better than BC. Meanwhile, BC-D, which learns the policy and level of demonstrators' expertise, also performs better than BC and is comparable to Co-teaching. Nonetheless, the performance of Co-teaching and BC-D is still much worse than VILD with IS.

**Accuracy of estimated expertise parameter.** Figure 9 shows the estimated parameters $\boldsymbol{\omega} = \{\mathbf{c}_k\}_{k=1}^{K}$ of $\mathcal{N}(\mathbf{u}_t|\mathbf{a}_t, \mathrm{diag}(\mathbf{c}_k))$ and the ground-truth variance $\{\sigma_k^2\}_{k=1}^{K}$ of the Gaussian noisy policy $\mathcal{N}(\mathbf{u}_t|\mathbf{a}_t, \sigma_k^2 \boldsymbol{I})$. The results show that VILD learns an accurate ranking of the variance compared to the ground-truth. The values of these parameters are also quite accurate compared to the ground truth, except for demonstrators with low-levels of expertise. A possible reason for this phenomena is that low-quality demonstrations are highly dissimilar, which makes learning the expertise more challenging. We can also see that the difference between expertise parameters of VILD with IS and VILD without IS is small and negligible.

**InfoGAIL with different values of context.** Figure 10 shows the learning curves of InfoGAIL across different values of context $z$. We can see that the performance of InfoGAIL depends on the context, i.e., there is a discrepancy between the best and worst performances of InfoGAIL. The discrepancy is clearer in the Walker2d task with the TSD noisy policy and in the robosuite reaching task (Figure 4).

**Robosuite reaching task.** Table 4 reports the performance in the last iterations in the robosuite reaching task experiments. It can be observed that VILD with IS outperforms comparison methods in terms of the mean performance.

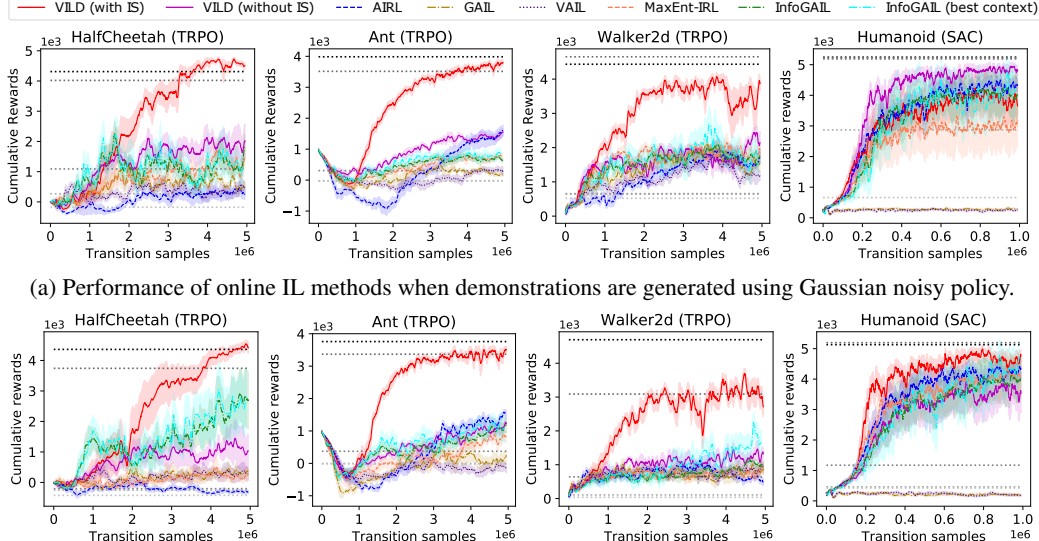

(a) Performance of online IL methods when demonstrations are generated using Gaussian noisy policy.

(b) Performance of online IL methods when demonstrations are generated using TSD noisy policy.

Figure 7: Performance averaged over 5 trials of online IL methods against the number of transition samples. Horizontal dotted lines indicate performance of $k = 1, 3, 5, 7, 10$ demonstrators.

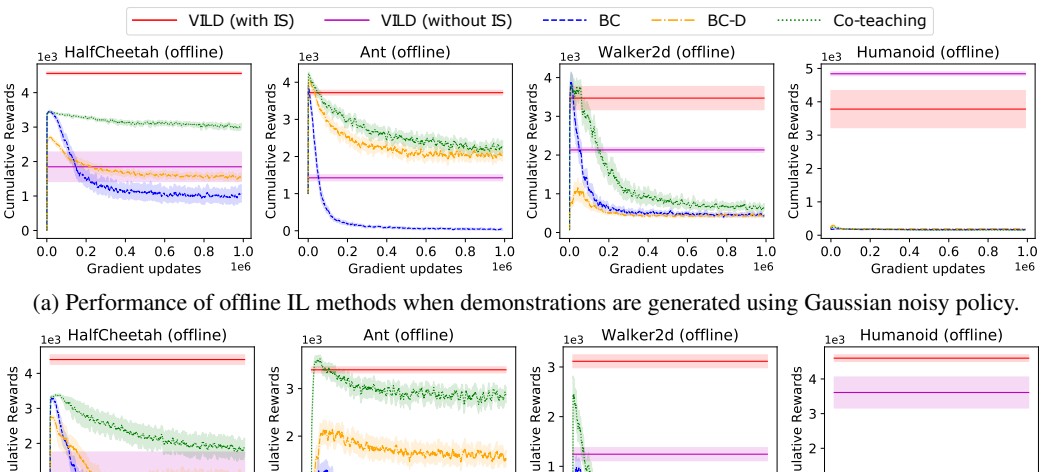

(a) Performance of offline IL methods when demonstrations are generated using Gaussian noisy policy.

(b) Performance of offline IL methods when demonstrations are generated using TSD noisy policy.

Figure 8: Performance averaged over 5 trials of offline IL methods against the number of gradient update steps. For VILD with and without IS, we report the final performance in Table 1.

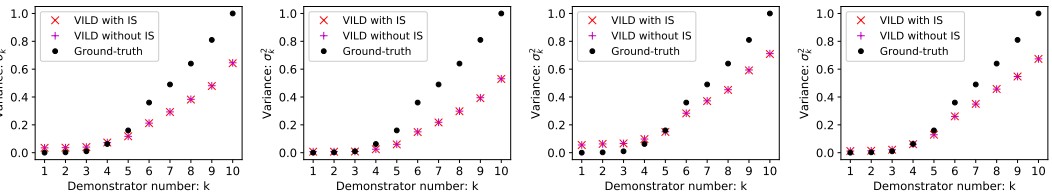

Figure 9: Expertise parameters $\boldsymbol{\omega} = \{\mathbf{c}_k\}_{k=1}^K$ learned by VILD and the ground-truth $\{\sigma_k^2\}_{k=1}^K$ for the Gaussian noisy policy. For VILD, we report the value of $\|\mathbf{c}_k\|_1/d_{\mathbf{a}}$.

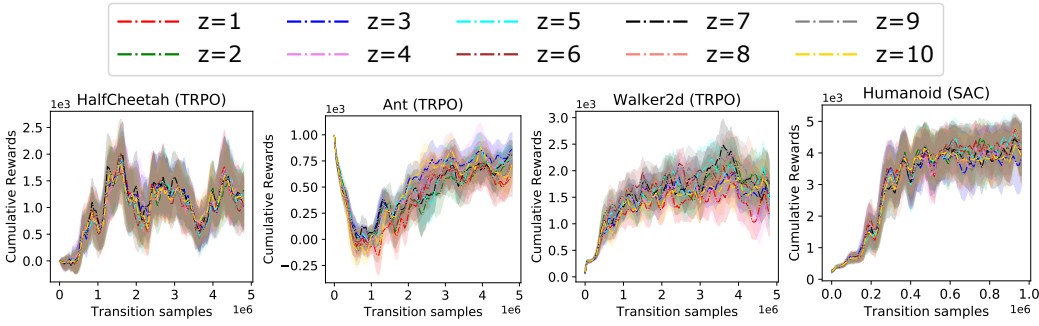

(a) Performance of InfoGAIL with different $z$ when demonstrations are generated using Gaussian noisy policy.

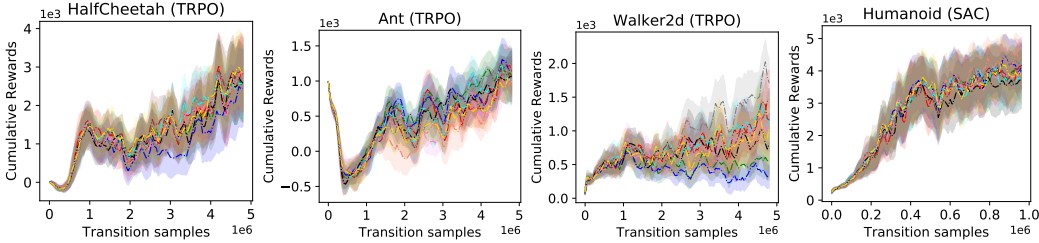

(b) Performance of InfoGAIL with different $z$ when demonstrations are generated using TSD noisy policy.

Figure 10: Performance averaged over 5 trials of InfoGAIL with different values of context $z$ for the benchmark tasks. For each trial, the performance of each context is computed using 10 test trajectories.

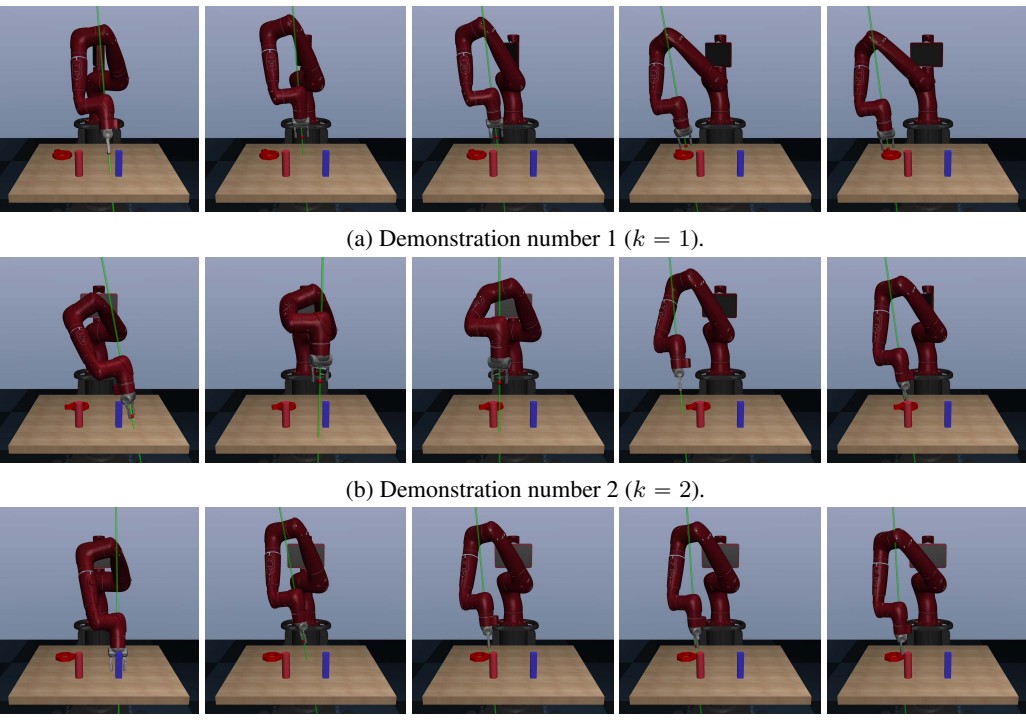

(a) Demonstration number 1 ($k = 1$).

(b) Demonstration number 2 ($k = 2$).

(c) Demonstration number 3 ($k = 3$).

Figure 11: Three examples of crowdsourced demonstrations in the robosuite reaching experiment.

