# OpenReview forum: "VILD: Variational Imitation Learning with Diverse-quality Demonstrations"
_ICLR.cc/2020/Conference — Reject_

### Official Review · AnonReviewer2 · 2019-10-23
**Official Blind Review #2**

**Rating:** 6

**Review:**

This paper proposes an imitation learning algorithm for the setting where the demonstration data consists of trajectories from sources of varying expertise. The authors proceed by defining a parameterized model of the (demonstration) trajectory distribution (Equation 2), which uses the MaxEnt-RL model for the optimal policy, and a distribution (p_w) to model the level of expertise. Imitation learning is then reduced to maximum-likelihood training under the provided demonstrations. Using appropriate variational distributions and model specification, the MLE objective is transformed to the VILD objective (Equation 5), which can be optimized with gradient descent. Expertise-level (p_w) is modeled as a Gaussian blur over the optimal action, wherein the variance is correlated with expertise (lower is better). Furthermore, a truncated IS approach is proposed for learning a better reward function. It samples more frequently from the experts that have a higher estimated expertise.

Overall, I really enjoyed reading the paper. The writing and the presentation of material (both background and novel solutions) is clean and concise. The Appendix, with all the derivations and the summarizations of the related approaches, is very informative. I would like the authors to comment on the following:

1.	It is claimed in Section 1 that prior approaches for imperfect imitation learning rely on auxiliary information from the expert, in the form of confidence scores or ranking, while VILD doesn’t use any. In my opinion, the fact that VILD uses “labeled” expert demonstrations (i.e. each demonstration is tagged with a number {1..K}) classifies as auxiliary information. Contrary to approaches such as InfoGAIL, which infer the latent structure of the expert demonstrations in a completely unsupervised fashion, VILD fixes the demonstrations-model and instead attempts to learn the parameters corresponding to this model – this holds exactly for the Gaussian policy, and approximately for the TSD policy setting.

2.	The difference in performance of VILD w/ and w/o IS is surprising. I understand the motivation in Section 3.4 that IS should help to improve the convergence rate, but for benchmarks like HalfCheetah, Walker, the performance seems to have saturated to a significantly lower value. I would like to know if the authors have some thoughts on this wide discrepancy w/ and w/o IS.

3.	Baselines – I’m not sure if InfoGAIL with a uniform prior on the context is a fair comparison to VILD-IS. Since VILD-IS changes the demonstrator sampling from uniform to expertise-dependent, one could do something similar for InfoGAIL – e.g. after training, report the best performing context, or sample context based on a performance-dependent distribution.

4.	Sample-efficiency in terms of expert data – Is the number of trajectories that are collected from each of 10 demonstrators reported somewhere?

**Experience Assessment:**

I have published one or two papers in this area.

**Review Assessment: Checking Correctness Of Derivations And Theory:**

I assessed the sensibility of the derivations and theory.

**Review Assessment: Checking Correctness Of Experiments:**

I assessed the sensibility of the experiments.

**Review Assessment: Thoroughness In Paper Reading:**

I read the paper thoroughly.

---

> ### Author Response · Authors · 2019-11-11
> **Reply to reviewer 2**
>
> Thank you for the reviews. Our responses to reviewer 2’s comments are provided below.
>
> 1. It is claimed in Section 1 that prior approaches for imperfect imitation learning rely on auxiliary information from the expert, in the form of confidence scores or ranking, while VILD doesn’t use any. In my opinion, the fact that VILD uses “labeled” expert demonstrations (i.e. each demonstration is tagged with a number {1..K}) classifies as auxiliary information.
> - Indeed, VILD requires the auxiliary numbers k \in {1..K} to be provided along with demonstrations. However, these numbers are not needed to be provided by experts. When k is not given, a reasonable strategy is to set k = n and K=N. In other words, we assume that there is a one-to-one mapping between demonstration and demonstrator. This is the strategy we used to set K=10 in the experiment with real-world data, where we have N=10 demonstrations but do not know the true number of demonstrators. The experimental results suggest that this strategy is reasonable, and that we do not need k to be provided by experts.
>
> 2. The difference in performance of VILD w/ and w/o IS is surprising. I understand the motivation in Section 3.4 that IS should help to improve the convergence rate, but for benchmarks like HalfCheetah, Walker, the performance seems to have saturated to a significantly lower value. I would like to know if the authors have some thoughts on this wide discrepancy w/ and w/o IS
> - A possible reason for the discrepancy in convergence is that the reward function in VILD with IS is “local” while the reward function in VILD without IS is “global”.  Specifically, VILD with IS learns the reward function from high-expertise demonstrators and agent’s policy, while VILD without IS learns the reward function from all demonstrators and agent’s policy. The support of state-action distribution induced by high-expertise demonstrators and agent’s policy is smaller than the support induced by all demonstrators and agent’s policy. Learning a function over small (i.e., local) supports is generally more data efficient than learning a function over large (i.e., global) support. Thus, VILD with IS is more data efficient than VILD without IS. The discrepancy in data efficiency can be reduced by using sample-efficient optimization methods such as SAC, as shown in the experiment with the humanoid task.
>
> 3. Baselines – I’m not sure if InfoGAIL with a uniform prior on the context is a fair comparison to VILD-IS. Since VILD-IS changes the demonstrator sampling from uniform to expertise-dependent, one could do something similar for InfoGAIL – e.g. after training, report the best performing context, or sample context based on a performance-dependent distribution.
> - As the reviewer suggested, we include InfoGAIL (best context) as a baseline in the revision, where we choose the best context variable for InfoGAIL according to the performance across all contexts (Figure 10 for benchmarks and Figure 4 for real-world data). As expected, using the best context improves the performance of InfoGAIL, especially for real-world data. Still, InfoGAIL (best context) and InfoGAIL (uniform distribution of contexts) are outperformed by VILD in terms of the final performance. Moreover, InfoGAIL (best context) is quite impractical since choosing the best context requires an expert to evaluate performance of all contexts.
>
> Using performance-dependent context distribution would also improve the performance of InfoGAIL. However, this approach is not applicable to our IL settings, since it requires access to performance evaluation metrics (e.g., ground-truth rewards or experts) during training. In contrast, the sampling distribution in VILD is based on the level of demonstrators’ expertise which is estimated without access to performance evaluation metrics.
>
> 4. Sample-efficiency in terms of expert data – Is the number of trajectories that are collected from each of 10 demonstrators reported somewhere?
> - In the benchmark experiments, each demonstrator collects trajectories with approximately 1000 time-steps. Thus, the total number of state-action pairs is approximately 10000 in benchmark experiments. In the experiment with real-world data, we use 10 demonstrations whose length are approximately 500 time-steps. Thus, the total number of state-action pairs is approximately 5000 in this experiment. We have clearly stated these numbers in the revision.
>
> Note that the number of state-action pairs is noticeably large compared to prior works. This is because we consider scenarios of large data with imperfect information. On the contrary, prior works consider scenarios of small data with (almost) perfect information. For this reason, we do not evaluate data-efficiency in terms of expert data in this paper, and only evaluate data-efficiency in terms of agent’s additional transition data.

---

### Official Review · AnonReviewer1 · 2019-10-24
**Official Blind Review #1**

**Rating:** 3

**Review:**

The paper considers imitation learning from a set of demonstrations with diverse-qualities. It proposes a graphical model describing the generation of these demonstrations and a variational approach for learning optimal policies from these demonstrations. The effectiveness of the approach is demonstrate on some continuous-control benchmarks on which they outperform other state of the art methods.

The paper addresses and interesting an important problem. However, although the experiments demonstrate good performance on a set of tasks they fail to provide convincing evidence about the generality of the approach. In particular, the model for generating diverse-quality demonstrations is tightly coupled to the optimal policy through the assumed demonstrations. This is also tightly coupled with the considered q functions. In practice, sub-optimal demonstrations are more likely to be generated from "experts" with different biases or wrong model assumptions and thus exhibit different patterns, and we might not know a good form for the posterior. From the current experiments it is unclear, whether the proposed approach would work in such cases.

Some more comments:
* I am missing some experimental details. For instance, how precisely is InfoGAIL used? Is the average performance when sampling from a uniform prior reported (as suggested by the paragraph in the experiments section)? If so, it would be interesting to also see the best performance over all contexts. Clearly, this could not be implemented be practice but could be facilitated in combination with an expert which can identify a good policy.
* What happens if the mismatch between expert and model becomes bigger? There is a hint in that direction for time dependent noise but additional insights would be welcome.
* Are there any theoretical insights into when the learning can work and when it can/will fail? In particular, when considering model mis-specification one can assume all kinds of worrying things happen.

**Experience Assessment:**

I have read many papers in this area.

**Review Assessment: Checking Correctness Of Derivations And Theory:**

I did not assess the derivations or theory.

**Review Assessment: Checking Correctness Of Experiments:**

I assessed the sensibility of the experiments.

**Review Assessment: Thoroughness In Paper Reading:**

I read the paper at least twice and used my best judgement in assessing the paper.

---

> ### Author Response · Authors · 2019-11-11
> **Reply to reviewer 1**
>
> Thank you for the reviews. Our responses to reviewer 1’s comments are provided below.
>
> *Although the experiments demonstrate good performance on a set of tasks they fail to provide convincing evidence about the generality of the approach.
> - We include an experiment with real-world data in the revision. The experimental results show that VILD (with IS) outperforms GAIL, AIRL, and MaxEnt-IRL. VILD also outperforms InfoGAIL in terms of the final performance. The results demonstrate that VILD is robust against real-world demonstrations, and that the choice of model in VILD is reasonable.
>
> *In practice, sub-optimal demonstrations are more likely to be generated from "experts" with different biases or wrong model assumptions and thus exhibit different patterns, and we might not know a good form for the posterior.
> - While sub-optimality in demonstrations may be caused by biased experts, we argue that different expertise of demonstrators also causes sub-optimality. This argument is supported by the experiment with real-world data, where VILD outperforms existing methods that do not consider expertise of demonstrators. VILD also outperforms InfoGAIL, which handles demonstrations from many (presumably biased) experts, in terms of the final performance.
>
> *How precisely is InfoGAIL used? Is the average performance when sampling from a uniform prior reported (as suggested by the paragraph in the experiments section)? If so, it would be interesting to also see the best performance over all contexts.
> - For InfoGAIL, we reported the performance averaged over a uniform distribution of contexts. As suggested, in the revision we include InfoGAIL (best context) in experiments, where we choose the best context variable for InfoGAIL according to the test performance across all contexts (Figure 10 for benchmarks and Figure 4 for real-world data). As expected, using the best context improves the performance of InfoGAIL, especially for real-world data. Still, InfoGAIL (best context) and InfoGAIL (uniform distribution of contexts) are outperformed by VILD in terms of the final performance. Moreover, InfoGAIL (best context) is quite impractical since choosing the best context requires an expert to evaluate performance of all contexts.
>
> *What happens if the mismatch between expert and model becomes bigger? There is a hint in that direction for time dependent noise but additional insights would be welcome.
> - The performance of VILD is expected to decrease as the mismatch between data and model becomes bigger. Nonetheless, the experimental results on time-dependent noise and real-world crowdsourced data suggest that the Gaussian model is quite robust against model mismatch.
> We note that model mismatch is an open issue in machine learning, since we generally do not exactly know the data distribution. This issue is often remedied by using more flexible models such as neural networks. While in this paper we consider a simple model where p_w is a Gaussian distribution with state-independent covariance C_w(k), we expect that VILD can be extended to more flexible models of p_w. We gave an example of such possibility in Eq. (14), where the covariance C_w(s,k) of Gaussian distribution depends on states and can be parameterized by neural networks. This model is a strict generalization of the one we used in experiments, since it enables modeling demonstrators whose level of expertise depends on states.
>
> Another approach to handle model mismatch is tempered posterior [1], which was shown to be effective in regression under the Bayesian inference setting. Applying this approach to VILD corresponds to scaling the reward function by a parameter \eta and learning \eta by the Bayesian inference approach. We leave an investigation of such an approach for handling model mismatch in VILD for future work.
>
> [1] Peter Grunwald and Thijs van Ommen. Inconsistency of Bayesian Inference for Misspecified Linear Models, and a Proposal for Repairing It. Bayesian Analysis, 2017.
>
> *Are there any theoretical insights into when the learning can work and when it can/will fail? In particular, when considering model mis-specification one can assume all kinds of worrying things happen.
> - We do not have any theoretical results right now. We do agree that such results are very useful and leave them for future work.

---

### Official Review · AnonReviewer3 · 2019-10-26
**Official Blind Review #3**

**Rating:** 6

**Review:**

I like this paper: it tackles an interesting and relevant problem (imitation learning when demonstrations come from people with different levels of expertise), takes the natural approach of attempting to infer which expert produced which demonstration, and shows results compared against a large number of baselines. However, I have some worries about whether VILD will work in more realistic settings, and so I’m only recommending a weak accept.

The experiments are done very well -- there are *many* baselines and a reasonable number of environments. However, the experiments are set up to match VILD’s model, and it is not as clear what would happen in a more realistic setting where there will be misspecification. For example, one hyperparameter of VILD is the assumed number of demonstrators, which is set to exactly the right number (10) in the experiments. I suspect that given the way the demonstrations are generated, it would be relatively easy to cluster the demonstrations into the 10 sets, making VILD’s job relatively easy. In contrast, with real data from humans, I expect that such a clustering would be much harder, since demonstrations from a single human often also have diverse quality. It remains to be seen how well VILD would perform in such a situation.

The authors do consider one type of misspecification: when instead of Gaussian noise, the true actions are generated with TSD noise. This gives me more hope that VILD will work in more realistic settings. While I would particularly appreciate experiments with real human data, in the absence of that I would like to see an experiment with misspecification of the number of demonstrators. For example, perhaps assume 5, 20 or 50 demonstrators, when there are exactly 10 demonstrators, and assume 10 demonstrators when there is actually just 1 demonstrator. Presumably VILD should not perform as well as e.g. GAIL in the latter case.

I was confused reading Sections 1 and 2. Prima facie, the model in Figure 1b is very strange: given that we have to model both p(at | st) and p(ut | at, st, k), it’s not clear why we even have an extra variable -- why couldn’t we just model p(at | st, k) directly? The answer is only made clear later in Section 3: we are specifically assuming that there is Gaussian noise on top of the chosen action. I would make this clearer in Section 2.

**Experience Assessment:**

I have read many papers in this area.

**Review Assessment: Checking Correctness Of Derivations And Theory:**

I did not assess the derivations or theory.

**Review Assessment: Checking Correctness Of Experiments:**

I assessed the sensibility of the experiments.

**Review Assessment: Thoroughness In Paper Reading:**

I read the paper at least twice and used my best judgement in assessing the paper.

---

> ### Author Response · Authors · 2019-11-11
> **Reply to reviewer 3**
>
> Thank you for the reviews. Our responses to reviewer 3’s comments are provided below.
>
> *However, the experiments are set up to match VILD’s model, and it is not as clear what would happen in a more realistic setting where there will be misspecification.
> - We include an experiment with real-world data in the revision. The experimental results show that VILD (with IS) outperforms GAIL, AIRL, and MaxEnt-IRL. VILD also outperforms InfoGAIL in terms of the final performance. The results demonstrate that VILD is robust against real-world demonstrations, and that the choice of model in VILD is reasonable.
>
> *One hyperparameter of VILD is the assumed number of demonstrators, which is set to exactly the right number (10) in the experiments.
> - When k is not given, a simple strategy is to set k = n and K=N. In other words, we assume that there is a one-to-one mapping between demonstration and demonstrator, and that one demonstration is collected by one demonstrator. This is the strategy we used to set K=10 in the experiment with real-world data where do not know the true number of demonstrators. The experimental results suggest that this strategy is reasonable.
>
> *I would like to see an experiment with misspecification of the number of demonstrators. For example, perhaps assume 5, 20 or 50 demonstrators, when there are exactly 10 demonstrators, and assume 10 demonstrators when there is actually just 1 demonstrator. Presumably VILD should not perform as well as e.g. GAIL in the latter case.
> - We expect the experiment with real-world demonstrations will address this concern regarding misspecification. Nonetheless, we agree that benchmark experiments on different values of K would shed more light onto this issue. We are running such experiments. However, we might not finish such experiments within the rebuttal period. We will include such experiments later.
>
> Regarding the assumption that K=10 when there is just 1 demonstrator, we expect VILD to perform comparable to standard IL methods and not worse. This is because VILD would learn p_w where all k’s yield equally small covariance. Since VILD do not encourages diversity among estimated expertise, the values of the covariance can be similar for all k’s.
>
> * I was confused reading Sections 1 and 2. Prima facie, the model in Figure 1b is very strange: given that we have to model both p(at | st) and p(ut | at, st, k), it’s not clear why we even have an extra variable -- why couldn’t we just model p(at | st, k) directly? The answer is only made clear later in Section 3: we are specifically assuming that there is Gaussian noise on top of the chosen action. I would make this clearer in Section 2.
> - We include a paragraph explaining the reasoning behind the model of data distribution in the revision (right after Eq. (1) in Section 2). We expect this paragraph to improve the clarity regarding our choice of model.

---

### Author Response · Authors · 2019-11-11
**Common reply to all reviewers**

Thank you for the reviews. In this reply, we describe changes in the revision, which address common concerns of reviewers. Briefly, we include an experiment with real-world demonstrations, as well as including InfoGAIL with the best context in experiments.

- A major concern of reviewers is the robustness of VILD against real-world demonstrations. To address this, in Section 5.2 of the revision, we present an experiment using real-world demonstrations publicly available from Mandlekar et al. (2018). These demonstrations are collected by crowdworkers with different levels of expertise, which makes these demonstrations suitable to evaluate our method. (We planned to include this experiment in the initial submission but could not finish it in time.)

The experimental results show that VILD (with IS) outperforms GAIL, AIRL, and MaxEnt-IRL. VILD also outperforms InfoGAIL in terms of the final performance. The results demonstrate that VILD is robust against real-world demonstrations, and that the choice of model in VILD is reasonable.

- Another concern of reviewers is the presentation of InfoGAIL’s performance. To address this, we include InfoGAIL (best context) in experiments, where we choose the best context variable for InfoGAIL according to the test performance across all contexts (Figure 10 for benchmarks and Figure 4 for real-world data). As expected, using the best context improves the performance of InfoGAIL, especially for real-world data. Still, InfoGAIL (best context) and InfoGAIL (uniform distribution of contexts) are outperformed by VILD in terms of the final performance. Moreover, InfoGAIL (best context) is quite impractical since choosing the best context requires an expert to evaluate performance of all contexts.

---

### Decision · Program_Chairs · 2019-12-19

**Decision:**

Reject

**Comment:**

The paper proposes a new imitation learning algorithm that explicitly models the quality of demonstrators.

All reviewers agreed that the problem and the approach were interesting, the paper well-written, and the experiments well-conducted. However, there was a shared concern about the applicability of the method to more realistic settings, in which the model generating the demonstrations does not fall under the assumptions of the method. The authors did add a real-world experiment during the rebuttal, but the reviewers were suspicious of the reported InfoGAIL performance and were not persuaded to change their assessment.

Following this discussion, I recommend rejection at this time, but it seems like a good paper and I encourage the authors to do a more careful validation experiment, and resubmit to a future venue.